# Dynamics of history-dependent perceptual judgment

I. Hachen [1,4], S. Reinartz[1,2,4], R. Brasselet[1,3], A. Stroligo[1] & M. E. Diamond [1✉]

Identical physical inputs do not always evoke identical percepts. To investigate the role of stimulus history in tactile perception, we designed a task in which rats had to judge each vibrissal vibration, in a long series, as strong or weak depending on its mean speed. After a low-speed stimulus (trial $n-1$), rats were more likely to report the next stimulus (trial $n$) as strong, and after a high-speed stimulus, they were more likely to report the next stimulus as weak, a repulsive effect that did not depend on choice or reward on trial $n-1$. This effect could be tracked over several preceding trials (i.e., $n-2$ and earlier) and was characterized by an exponential decay function, reflecting a trial-by-trial incorporation of sensory history. Surprisingly, the influence of trial $n-1$ strengthened as the time interval between $n-1$ and $n$ grew. Human subjects receiving fingertip vibrations showed these same key findings. We are able to account for the repulsive stimulus history effect, and its detailed time scale, through a single-parameter model, wherein each new stimulus gradually updates the subject's decision criterion. This model points to mechanisms underlying how the past affects the ongoing subjective experience.

[1] Tactile Perception and Learning Laboratory, International School for Advanced Studies (SISSA), 34136 Trieste, Italy. [2]Present address: Brain & Sound Lab, Department of Biomedicine, Basel University, 4056 Basel, Switzerland. [3]Present address: Time Perception Lab, International School for Advanced Studies (SISSA), 34136 Trieste, Italy. [4]These authors contributed equally: I. Hachen, S. Reinartz. ✉email: diamond@sissa.it

Introspection tells us that a given sensory event may be experienced differently at different times, even if the physical input is replicated. This lack of constancy can be studied in the laboratory, where the identical physical input can yield different perceptual judgments across the extended sequence of trials typical of a controlled experiment. Paralleling the variability in subjective estimation and decision making, variability in the face of an unchanging physical input is also found in neuronal responses at all levels of the sensory pathway and at all levels of resolution, from single neurons to EEG[1–5].

Some studies have been able to link the variability in the judgment of physical input to the sequence of preceding trials—the trial history[6–15]. But trial history is itself complex and multifactorial, for it consists of previous stimuli, previous choices (expressed as actions), and previous outcomes (rewards, collected or lost). Recent literature has focused on dissociating the effect of past stimuli from the effect of the subject's own past choices[16–18]. When compared directly within the same experiment, the most recent choice (on trial $n - 1$) is found to exert an attractive effect on the current choice (trial $n$); that is, choices tend to be repeated[8,16,18,19]. This phenomenon has been argued to reflect high-level decisional processes[16,18]. By contrast, the most recent stimulus (trial $n - 1$) is found to exert a repulsive effect on the next reported percept (trial $n$). For instance, a vertical line or grating is judged as slightly tilted to the left if the previously viewed one was tilted to the right[16,18,20,21]. Perceptual repulsion has been argued to originate before the stage of decision making, being consistent with the phenomenon of sensory adaptation seen at many levels of the ascending sensory pathways[22], and occurs retinotopically[17] and independently of the requirements of the task[16]. Since even passive exposure to sensory input can cause a repulsive bias in the subsequent perceptual judgment[18], and since the repulsive effect in trial $n$ has been observed mostly after brief intervals (even below 1 s)[16,18], neuronal short-term adaptation has been proposed as a mechanism[23].

Whether the relevant temporal metric for the repulsion exercised by past trials is continuous time or the number of elapsed trials is an open question that must be resolved before characterizing an underlying mechanism. Repulsive effects have been detected[21,24,25] for conditions in which the stimulus exposure in the preceding trial ($n - 1$) is long compared to the time between trials ($n - 1$ to $n$). In the current study, we examine the $n - 1$ effects across time spans that are much longer than the single stimulus exposures. Our goal is to track the effect of a single stimulus across an extended string of trials and, further, across a densely sampled range of inter-trial durations. We trained rats to judge the vibrissal vibration presented on each trial as belonging to one of two categories, "strong" or "weak," according to a boundary set by the experimenter. By grouping trials according to three factors—(i) stimulus, (ii) choice, and (iii) reward—we are able to isolate stimulus-driven sequential effects. Next, we quantified the influence on trial $n$ of, not only the most recent stimulus ($n - 1$), but also stimuli extending farther back ($n - 2$ and earlier). By means of a model wherein each new stimulus causes a gradual updating of the rat's decision criterion that is played out on the next trial, we can account for the repulsive stimulus history effect and its detailed timescale. Using a single free parameter —the time course of the updating of the decision criterion—this model predicts rats' choices significantly better than a model based on the current stimulus alone. In psychophysics experiments, we extend these key observations to human subjects, arguing for the model's generality as an underlying mechanism to explain how the past affects the present.

## Results

**Categorization of vibration speed.** To investigate vibration perception and the effects of stimulus history, we designed a task in which rats had to judge the presented vibration as belonging to one of two categories, high-speed or low-speed (in shorthand, "strong" or "weak"). In Fig. 1a, trial structure is illustrated (upper tier) along with an example stimulus (middle tier). The rat initiated the trial (lower tier) by placing its snout in the nose poke. After a 400-ms delay, a vibration of 500 ms duration was delivered to the whiskers by a moving plate. Each vibration consisted of a sequence of speed values sampled from a half-Gaussian distribution (as in[26,27]; see Methods). A single vibration was thus defined by its nominal mean speed, denoted $sp$. There were nine Gaussian distributions, yielding nine possible $sp$ values; the probability, per trial, of selecting a given $sp$ value was uniform (11.1%), except where stated otherwise. An auditory cue, presented 400–600 ms after termination of the vibration, instructed the rat to withdraw from the nose poke and choose between the two reward spouts. Either the left or right spout was rewarded, according to whether vibration $sp$ was higher or lower than the category boundary. Correct choices were rewarded by liquid delivery (pear juice diluted in water) and accompanied by a reinforcing sound. For stimuli on the category boundary, reward was delivered with 0.5 probability for either choice. After a correct choice, the next trial could be initiated immediately by another nose poke; after an incorrect choice, reward was not delivered and the rat had to wait 2–4 s before initiating the next trial. The elapsed time between successive nose pokes was taken as the inter-trial interval (ITI; see Fig. 1a).

To correctly classify stimuli, rats needed to create an internal reference or apply a decision criterion, based on their previous experience of stimulus/choice/reward contingencies. Their performance was captured through logistic psychometric functions that assess the probability with which the stimulus was judged "strong" in relation to $sp$ (see Methods). A logistic function is illustrated schematically in Fig. 1b in black; it is dependent on the terms $\lambda$ (upper lapse rate), $\gamma$ (lower lapse rate), $\sigma$ (the standard deviation of the underlying cumulative Gaussian distribution), and $\mu$ (the stimulus value aligned to the inflection point). In this form of logistic function, the inflection point is equivalent to the mean of the underlying Gaussian distribution and we refer to $\mu$ as curve midpoint. The slope of the psychometric function at this point (gray) is proportional to $1/\sigma$. The point of subjective equality (PSE) is defined as the stimulus value at which the two choices are equally likely, i.e., $p(\text{"strong"}) = 0.5$. In the case of symmetric lapses ($\lambda = \gamma$), the PSE is equal to $\mu$.

Once trained, rats classified vibrations with an accuracy that increased with the stimulus distance from the category boundary. Figure 1c shows the psychometric functions for six rats, fitted to their behavioral data (points), averaged over 40–45 sessions per rat. The triangles on the upper and lower abscissa depict the curve midpoints for each rat.

**Classification is flexible.** How does the psychometric curve evolve over the course of a session? To assess performance at the outset of the session, we fit a function separately using only the first three trials of all sessions, averaged across subjects; we fit another psychometric function using three randomly selected trials per session (Fig. 2a, light blue and dark blue, respectively). The category boundary was 103 mm/s. Acuity was poor at session onset, as quantified by the maximum slope of the psychometric curve (ordinate of right panel; within-session improvement is shown in greater detail in Supplementary Fig. S1a). The early-session performance exhibited no overall bias toward judging stimuli in one or the other category, as revealed by the PSE symmetry about 103 mm/s (abscissa of right panel).

We asked how rats would adjust their choices in response to changes in range and category boundary. For a subset ($n = 2$) of

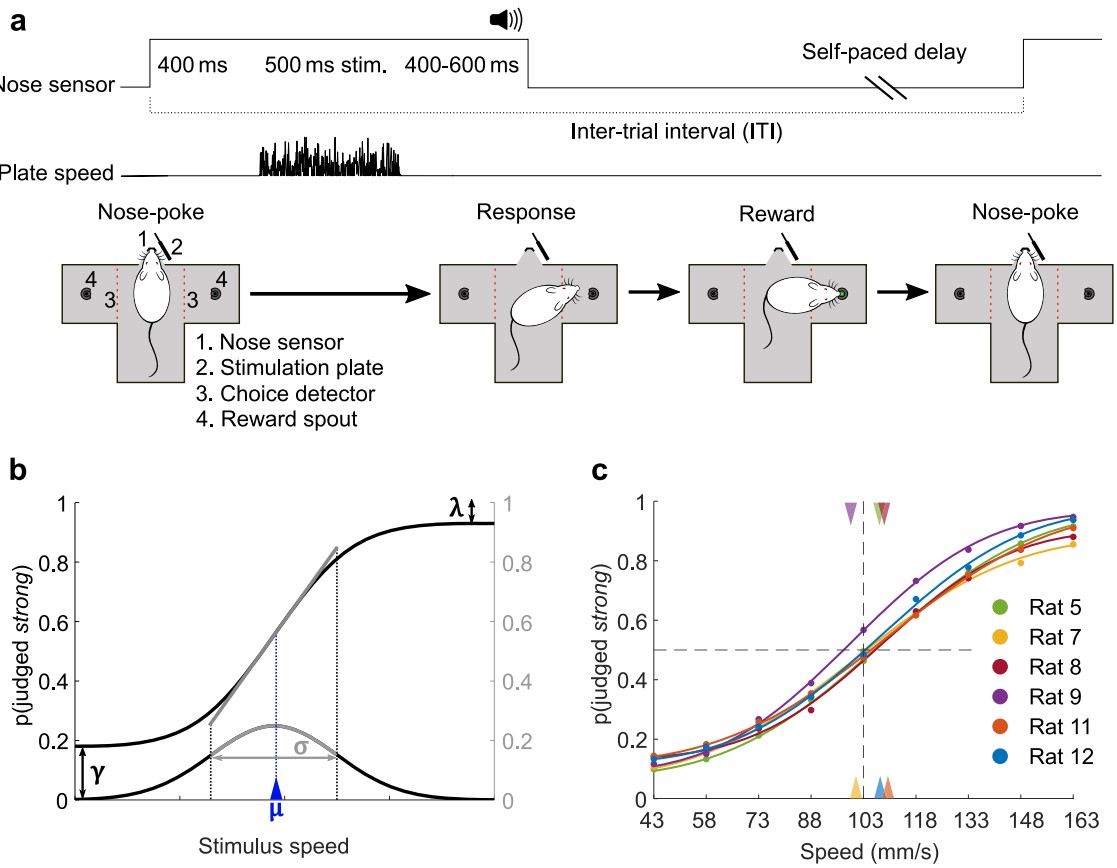

**Fig. 1 Vibrotactile categorization task and overall behavioral data. a** Trial configuration. By placing its snout in the nose poke, the rat triggered a whisker vibration. The rat withdrew upon the go cue, and its turn direction was detected by an optic sensor (red dashed line). **b** Example psychometric curve, generated using a cumulative Gaussian distribution function with lapse parameters $\gamma$ and $\lambda$. Slope at inflection point is highlighted (gray segment). **c** Psychometric functions fitted to the averaged data of six rats (approximately 15,000 trials per rat). Triangles mark curve midpoints. Source data are provided as a Source Data file.

the rats who already performed stimulus categorization with a stable range and boundary (Fig. 1c), we proceeded to vary the stimulus set from session to session, alternating between low and high ranges with commensurate shifts in the category boundary (Fig. 2b). The lowest *sp* of the high-range stimulus set (43 mm/s) corresponded to the second lowest *sp* of the low range. Analogously, the highest *sp* of the low-range stimulus set (148 mm/s) corresponded to the second-highest *sp* of the high range. The category boundary also shifted by one step, from 88 mm/s (low range) to 103 mm/s (high range). Rats were tested daily for 6 weeks with alternation every weekday between the two ranges. We expected them to detect the current session's rule (similarly to[28], however without cues and without explicit training for the shift). Of particular interest was whether at the beginning of a session they would call up the decision boundary of the previous session, or else form the new boundary ex novo.

The whole-session psychometric curves associated with the two stimulus ranges are shown in Fig. 2b, averaged across sessions and rats. The shift in the psychometric curve PSE, with no accompanying change in slope, reflects the rats' ability to conform their behavior to the current stimulus range and/or boundary (see inset).

To evaluate the time course of this adjustment, Fig. 2c shows the average psychometric curves of rats across the first three trials of every session, yielding two main findings. First, at the session onset the rats showed low performance (shallow logistic functions and high lapse rates), consistent with the results of Fig. 2a. Second, the absence of significant separation between the two

curves suggests that the psychometric function of the preceding session was not carried over to bias choices in the new session. If it were carried over, the blue curve would have been displaced to the left of the light green curve, inasmuch as the current low-range session would initiate with a choice function from the preceding high-range session, and vice versa. Further analyses (Supplementary Fig. S1b) showed that the separation between the curves became statistically significant after about 30 trials (rank-sum test between PSEs, $p < 0.05$; after about 50 trials, $p < 0.001$). In conclusion, rats initiated the session with poor performance and without any observable residual influence of the previous session's range and/or boundary; their behavior adapted over time to the current session.

**Are psychometric curves modulated by the boundary or by the stimulus distribution?** The difference between light green versus blue curves in Fig. 2b, c could result from the rats matching their own decision criterion to the current session's category boundary. Alternatively, rats may not form a decision criterion explicitly; they may instead be influenced only by the stimulus range and the extraction probability of each *sp* value within that range. In both the low- and the high-range sessions, the category boundary lay exactly at the center of the range, such that "splitting" the presented stimulus range at its midpoint would give the appearance of aligning choices to the boundary. To select between the alternatives, we designed two stimulus sets with boundary at either 88 mm/s (light green) or 118 mm/s (blue), both of them off-center with respect to the range (Fig. 2d, e). Range itself was

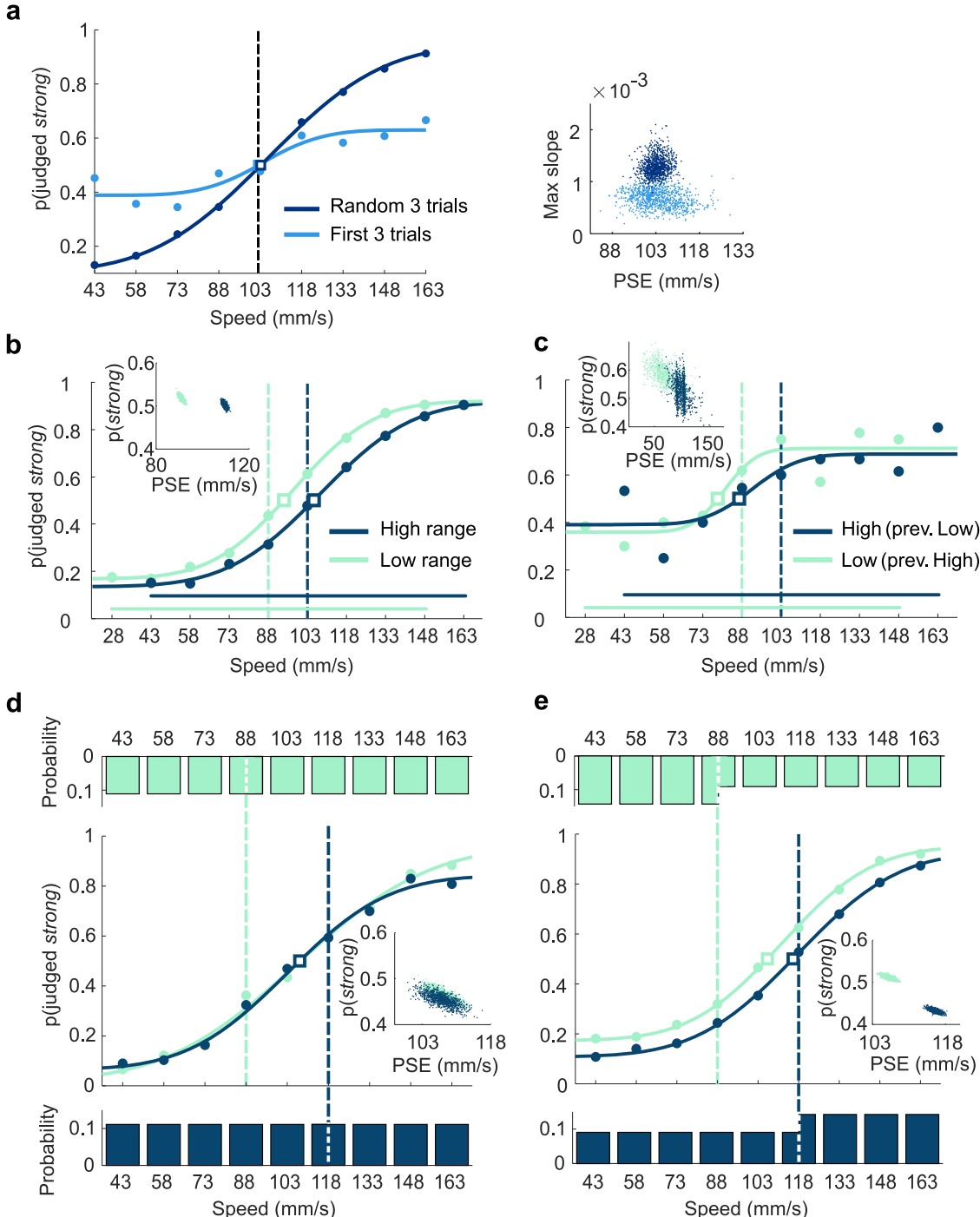

**Fig. 2 Rats flexibly adapt to stimulus range during each session. a** Psychometric curves for the first three trials of every session (light blue), and three trials randomly sampled from each session (dark blue). Stimulus range and boundary was the same as in Fig. 1c. Category boundary is illustrated as a dashed vertical line and point of subjective equality (PSE) as a square. Scatter plot shows the bootstrap of PSE and maximum slope from the same data set. **b** Psychometric curves in the "low-range" (light green) and the "high-range" (blue) sessions. Scatterplots show the bootstrap of PSE and average probability of "strong" judgment (excluding stimulus values of 28 and 163 mm/s) estimated from the same data set. Data analyzed by grouping together trials from two different rats with opposite reward rule. **c** Psychometric curves obtained by averaging the first three trials of all "low-range" (light green) and "high-range" (blue) sessions. Scatterplots show the bootstrap of PSE and average probability of "strong" judgment (excluding stimulus values of 28 and 163 mm/s) estimated from the same data set. **d** Psychometric curves in the Boundary 88 (light green) and the Boundary 118 (blue) sessions with equal ranges and with extraction probability set uniformly across stimulus values. Histograms in **d** and **e** depict the stimulus extraction probabilities. Scatterplots show the bootstrap of PSE and average probability of "strong" judgment estimated from the same data set. **e** Psychometric curves in the Boundary 88 (light green) and the Boundary 118 (blue) sessions with equal ranges and with extraction probability held equal for the two categories. Source data are provided as a Source Data file.

invariant. In one paradigm, the probability of each $sp$ value was held uniform at 0.11 (Fig. 2d, upper and lower histograms). This means that with the boundary at 118 mm/s, the stimuli of the "strong" category were presented less frequently (7/18 of all trials) than were the stimuli of the "weak" category (11/18 of all trials). With the boundary at 88 mm/s, the "strong" and "weak" proportions were reversed.

The second paradigm replicated the first except that the extraction probability of each $sp$ value was adjusted to yield an equal number of presentations from the "weak" and "strong" categories (Fig. 2e, upper and lower histograms). In alternating sessions, the boundary was moved to the left (light green) and to the right (blue). Two rats were tested in each paradigm; one rat participated in both paradigms and two rats participated in just one paradigm. When the category boundary was shifted and $sp$ probabilities were held uniform at 0.11, the psychometric functions were overlapping (Fig. 2d; see inset for quantification) —the observed decision criterion was insensitive to the session's category boundary. In contrast, when the category boundary was shifted and $sp$ probabilities were altered to equalize category probability, the psychometric functions were well separated (Fig. 2e, see inset for quantification). The results suggest that choices were dictated by stimulus distribution but not by boundary—the alignment of the rat's decision criterion to the session's category boundary, evident both in Fig. 2b and in Fig. 2e, seems to be an outcome of the stimulus distribution to which the rat was exposed, not the reward rule boundary or the frequency of reward associated with a given category. There likely exist experimental and real-life conditions in which perceptual choices are determined as much by reward rule as by stimulus distribution; however, the present conditions seem to cause rats to link their decisions to the statistical structure of the stimulus set. Such stimulus-dependent decision making motivates a detailed analyses of how past stimulus values generate current choices.

**Local stimulus history influences choice**. In the remainder of the study we focus on the influence of recent stimulus history. In all further analyses, the stimulus range was fixed (43–163 mm/s range, as in Fig. 1c) and the extraction probability of each $sp$ value was uniform. For simplicity, we denote the distance between stimulus $sp$ and the category boundary as $\Delta$speed, on a scale from $-4$ (the lowest $sp$) to 4 (the highest $sp$). According to the sign of $\Delta$speed, either the left or right spout was rewarded as correct. Figure 3a shows the psychometric curves for trial $n$, for all rats merged, sorted according to $\Delta$speed of trial $n-1$. This sorting resulted in a spread across the curves—when trial $n-1$ presented a high $\Delta$speed stimulus, the probability of trial $n$ being categorized "strong" decreased; when trial $n-1$ presented a low $\Delta$speed stimulus, the probability of trial $n$ being categorized "strong" increased. The effect was graded for intermediate values of trial $n-1$ $\Delta$speed. The PSE computed with all trials merged, unconditional on $n-1$ $\Delta$speed, is given by the black dashed vertical line.

Since $\Delta$speed and the rat's choice are correlated, we sought to determine whether the influence of trial $n-1$ on trial $n$ choice derived from the $n-1$ stimulus or the $n-1$ choice. The trial $n-1$ choice might cause a bias toward repeating the action in trial $n$ ("sticking") or a bias toward changing the action in trial $n$ ("switching"). Since stimuli closer to the extremes ($\Delta$speed = 4 and $-4$) yielded greater percentages of correct responses, the graded effect of the preceding trial's $\Delta$speed could still derive from an effect of previous choice. To test for this, we generated psychometric curves after excluding previous trials in which the animal made an incorrect choice; the results (Supplementary

Fig. S2) were identical to those of Fig. 3a. To further exclude such effects, we considered all trials in which $n-1$ $\Delta$speed = 0 and choice was rewarded; from this set, we extracted an equal number of $n-1$ trials with "weak" and "strong" judgments. We then plotted the trial $n$ psychometric curve according to whether trial $n-1$ was judged as "weak" or "strong" (Fig. 3b). If trial $n$ choice were related to the preceding action, the two curves would be significantly separated. Though there may be a trend for trial $n-1$ choice to affect the subsequent choice when trial $n$ stimuli were weakest ($\Delta$speed $-2$ to $-4$), curve parameters were not statistically different (bootstrap test, 90% confidence level; see Supplementary Fig. S3a). Instead, the overlap in confidence intervals argues that the trial $n$ choice was influenced only mildly, or not at all, by trial $n-1$ action or choice.

Reward history is another factor that might influence choice. Figure 3c shows the probability of trial $n$ being judged "strong" as a function of trial $n-1$ $\Delta$speed, with data separated according to whether the trial $n-1$ choice was correct (green) or incorrect (violet). All $\Delta$speed values of trial $n$ are pooled. Absence of any effect of trial $n-1$ $\Delta$speed per se would lead to flat curves. Furthermore, if the bias were driven by the previous choice rather than the previous stimulus, the violet curve would show a positive slope, opposite to that of the green curve. Instead, the negative slope of both curves follows from the "repulsive" effect of trial $n-1$ $\Delta$speed: the greater $n-1$ $\Delta$speed, the less likely for $n$ to be categorized as "strong." However, to better understand the small difference between the green and violet curves, we performed additional analyses of the potential role of reward history (Supplementary Table T1 and Supplementary Fig. S3b). A logistic regression model including an interaction term between previous stimulus and previous reward was fitted to the data. The interaction weight was statistically significant ($p < 0.001$; see Supplementary Table T1), indicating that the repulsive effect of stimulus $n-1$ was slightly reduced if that trial's choice was not rewarded (see also Supplementary Fig. S4). Bootstrapping the parameters of the psychometric curves from Fig. 3c (see Supplementary Fig. S3b) revealed this phenomenon to be mainly driven by an increase in lapse probability following an incorrect trial, suggesting a slight lose-switch tendency that might partly counterbalance the effect of previous stimulus after incorrect trials. The absence of a significant overlap (at the 90% confidence level; Supplementary Fig. S3b) in the slope parameters of the two curves emphasized again that the stimulus-driven bias seen in trial $n$ is mostly independent of the outcome (rewarded or not rewarded) of trial $n-1$. The repulsive effect of the preceding trial, under our experimental conditions, thus appears to depend primarily on the sensory-perceptual processing of the vibration itself, while the effects of $n-1$ choice (Fig. 3b) or reward (Fig. 3c) appear considerably less substantial.

What is the nature of the psychometric curve shift evoked by the preceding trial? In general, a psychometric curve may shift horizontally, which would have a larger effect on choices when $\Delta$speed is near 0 than on choices for stimuli close to $\Delta$speed of $-4$ and 4. A curve may shift vertically, which would affect choices uniformly across the full range of $\Delta$speed. To select between the two possibilities, we compared the psychometric curve parameters of a grand average psychometric curve of all rats, grouped according to $\Delta$speed in trial $n-1$ (as in Fig. 3a). Changes in PSE had a stronger linear relation with the horizontal midpoint parameter, $\mu$ ($R^2 = 0.7422$) than with the vertical lapse parameters, $\gamma - \lambda$ ($R^2 = 0.2242$). Additional results consistent with a largely horizontal curve shift are shown in Supplementary Fig. S5.

Given the mainly horizontal shift induced by stimulus $n-1$, the PSE constitutes a robust measure of the history-dependent bias. On the data set averaged across rats, nine values of the PSE associated with nine values of $n-1$ $\Delta$speed are shown as the blue

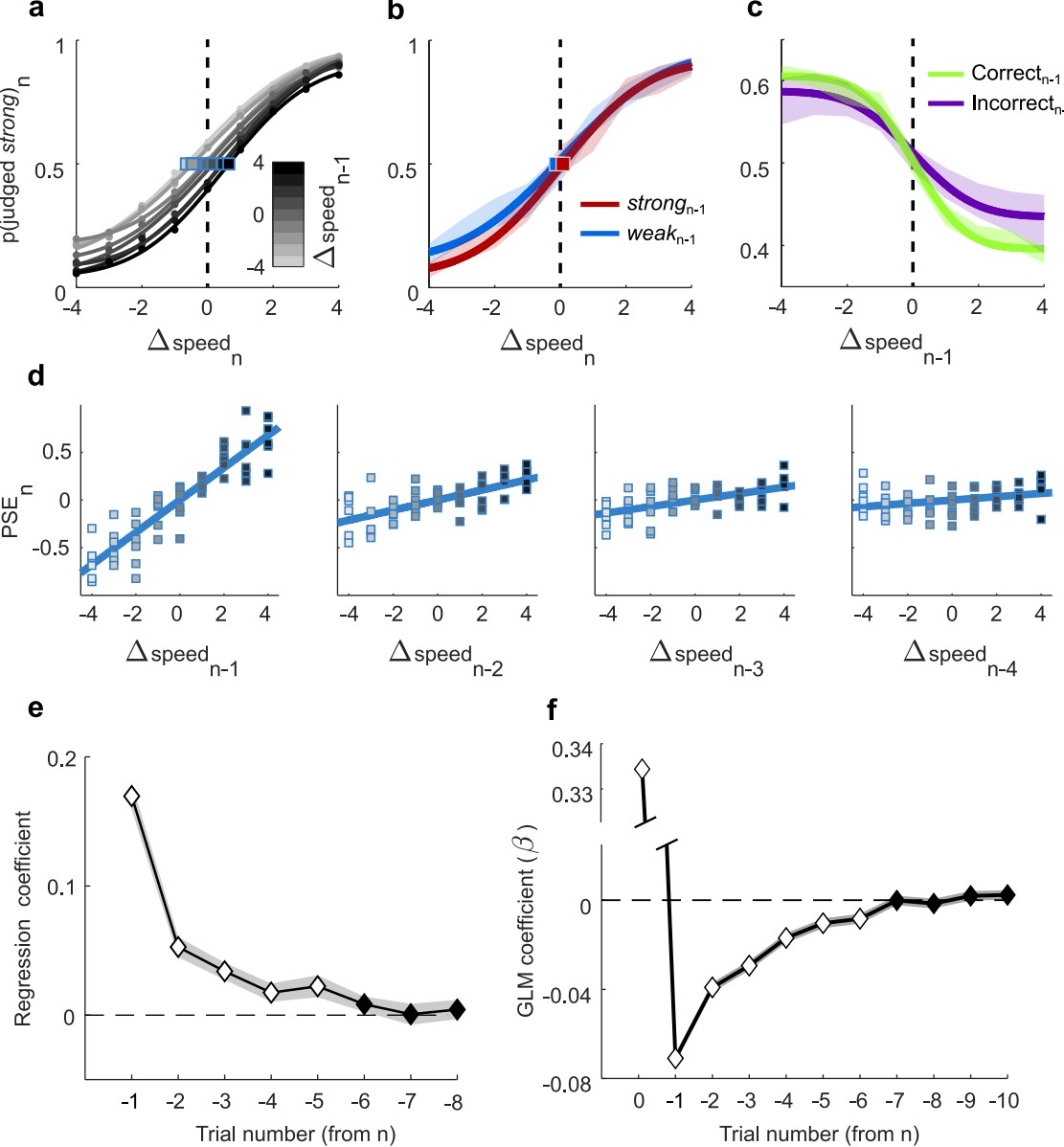

**Fig. 3 Trial-by-trial stimulus-dependent repulsive bias. a** Probability of categorizing stimulus $n$ as "strong" as a function of trial $n$ $\Delta$speed, with curves grouped by $\Delta$speed of trial $n-1$. Darker curves correspond to higher $n-1$ $\Delta$speed. Blue squares denote PSE. **b** Trial $n$ psychometric curves, grouped by trial $n-1$ choice, "strong" (red) and "weak" (blue). Transparent shading represents 95% confidence interval of bootstrapped probabilities. **c** Probability of categorizing stimulus $n$ as "strong" as a function of the $\Delta$speed of trial $n-1$. All $\Delta$speed values of trial $n$ are pooled. The green line corresponds to $n-1$ correct trials, the violet line to $n-1$ incorrect trials. Transparent shading represents 95% confidence interval of bootstrapped probabilities. **d** Bias of the trial $n$ psychometric curve, depending on $\Delta$speed in trial $n-1$ (far left plot) to trial $n-4$ (far right plot). Shading inside the squares denotes $\Delta$speed in trial $n-1$. Squares correspond to individual rats. **e** Slope of the regressions shown in panel **d**, in the eight preceding trials. White denotes slopes different from 0 ($p < 0.05$, two-tailed $t$-test). Transparent shading represents standard deviation between rats. **f** Coefficient values ($\beta$) of the GLM including stimuli of up to 10 preceding trials as predictors. White denotes coefficients different from 0 ($p < 0.05$, two-tailed $t$-test). Discontinuous ordinate shows large positive coefficient associated with stimulus $n$. Transparent shading represents standard error of the mean. Source data are provided as a Source Data file.

squares in Fig. 3a. These can be considered graded quantities of history-dependent curve shift, or bias. When the same procedure is carried out for six individual rats, six data points are associated with each value of $\Delta$speed on trial $n-1$ (Fig. 3d, left panel). The size of the trial $n$ bias was linearly correlated with trial $n-1$ $\Delta$speed (Pearson's $R = 0.869$; $p < 0.001$). This linear correlation quantifies the $n-1$-dependent curve shifts seen in Fig. 3a.

We then asked whether the stimuli in trials preceding $n-1$ affected the behavior on trial $n$. The next three plots of Fig. 3d (left to right) show trial $n$ bias in relation to $\Delta$speed on trials $n-2$, $n-3$, and $n-4$. Progressively more distant trials evoked a

progressively less pronounced bias, reflected in the decreasing slope of the regression lines (see also Fig. 3e).

To visualize more completely the trial-by-trial decay in the effect of more remote trials, Fig. 3e plots the slope of previous trial $\Delta$speed versus PSE shift (as in panel 3d), averaged across all sessions of all rats, for up to eight trials before $n$. The decay seems to follow an exponential trend, with a small but significant effect (white diamonds, $p < 0.05$) up to the fifth previous trial.

We further investigated how preceding stimuli contribute to choice by applying a logistic regression model, a more accurate method than the bias slopes of Fig. 3e. Specifically, we predict the

trial $n$ decision by the probit link function:

$$p(\text{strong})_n = \text{probit}^{-1}(\text{seq}_n) \qquad (1)$$

where

$$\text{seq}_n = \beta_0 + \beta_1 \Delta \text{speed}_n + \beta_2 \Delta \text{speed}_{n-1} + \ldots + \beta_i \Delta \text{speed}_{n-i} \qquad (2)$$

The term $\text{seq}_n$ represents the linear sum of the current stimulus, $\Delta \text{speed}_n$, and a history of up to $i$ preceding stimuli, each weighted with a coefficient $\beta_i$ while $\beta_0$ is an intercept term. Figure 3f shows that up to six preceding trials significantly (white diamonds, $p < 0.05$) affected choice on trial $n$. The weights appear to decay with an exponential trend similar to that obtained by estimating each preceding trial's individual contribution to the trial $n$ psychometric curve (Fig. 3e).

**Trial-by-trial discretized model of history-dependent choices.** The results reported in Fig. 2 indicate that choices were driven by the session's stimulus distribution, while those in Fig. 3 suggest that the most recent stimuli have the strongest influence on the current choice. Taken together, the findings suggest a model where choices are the outcome of comparing the percept evoked on trial $n$ to a decision criterion—a criterion set not by the investigator's reward rule (Fig. 2d) but by a weighted combination of preceding stimulus values. In this scenario, stimulus $n-1$ biases the trial $n$ choice by virtue of its "pull" on the decision criterion. Earlier stimuli exert a progressively less powerful pull.

The psychometric curve midpoint $\mu$, in our data set, is nearly equivalent to the PSE (see Supplementary Fig. S5), which divides the stimulus dimension into one range where $\Delta$speed is more likely to be judged as "weak" and the complementary range where $\Delta$speed is more likely to be judged as "strong." We therefore take $\mu$ on trial $n$ as a proxy for the rat's decision criterion, a model grounded in signal detection theory[29,30]. Since the bias carried over from past stimuli decayed exponentially, we formulated a recursive model of the criterion on trial $n$, $\mu_n$, as an exponentially weighted average of the history of stimuli up to and including $n-1$:

$$\mu_n = \mu_{n-1} \cdot e^{\left(-\frac{1}{\tau}\right)} + \Delta \text{speed}_{n-1} \cdot \left(1 - e^{\left(-\frac{1}{\tau}\right)}\right) \qquad (3)$$

$\Delta \text{speed}_{n-1}$ and $\mu_{n-1}$ (previous criterion, applied on trial $n-1$) are summed with a relative exponential weight given by the time constant $\tau$. In this model the only free parameter is $\tau$, providing a timescale (expressed in units of trial number) for the influence of preceding trials. (Although such models are often written with a parameter $\alpha = 1/\tau$, as for example[31], we chose this natural temporal representation for reasons that will become clearer in the next sections.) The rat's decision is a binary choice made by comparing its percept of $\Delta \text{speed}_n$ to the criterion $\mu_n$. For an example sequence of stimuli, Fig. 4a shows the trial-by-trial value of the computed decision criterion and illustrates how the model can predict some choices of the rat. Circles represent presented $\Delta$speed values. The dark area covers $\Delta$speed values rewarded as "strong" while the light area covers $\Delta$speed values rewarded as "weak"; $\Delta$speed of 0 was rewarded randomly. On trials denoted by filled and unfilled circles, the rat judged the stimulus as "strong" and "weak," respectively. The blue line depicts $\mu_n$ generated by the model of Eq. (3) and the blue squares highlight the $\mu_n$ value at the moment of stimulus presentation. Note the correct predictions on the two trials with $\Delta \text{speed} = 0$ (green asterisks). Because the criterion steps to a new value after each stimulus, we term this the discretized history model, in contrast to a subsequent continuous model.

In order to capture the graded effect of previous stimulus on current choices shown in Fig. 3a, we modeled the probability of

the choice "strong" on trial $n$ by employing the same logistic function used to fit psychometric curves (see Methods):

$$p(\text{strong})_n = \gamma + (1 - \gamma - \lambda) \cdot \frac{1}{2}\left[1 + \text{erf}\left(\frac{\Delta \text{speed}_n - \mu_n}{\sigma \sqrt{2}}\right)\right] \qquad (4)$$

where $\mu_n$ is updated trial-by-trial as the exponentially weighted average of past stimuli (Eq. (3)). The other three parameters of the function ($\gamma, \lambda, \sigma$) are fitted on the entire data set for each rat, leaving $\mu$ as the only history-dependent parameter.

To evaluate the model, we performed a cross-validation by repeatedly partitioning the sessions into two sets, one used for setting the time constant $\tau$ (80% of sessions), and one for testing model performance (20% of sessions). In this test of the discretized history model, the predicted choice is not set by the criterion as a binary divider, but as a smooth function, centered on $\mu_n$, which mirrors the history-dependent logistic function (Fig. 4b). While the model predicts choice on a given trial by aligning a smooth logistic function, the rat's decision is, of course, binary. To compute the prediction error, we employed the Brier score[32] (see Methods). The Brier score on a single trial can range from 0, where the logistic function matches the choice, to 1 where the logistic function value is at the opposite extreme of the observed choice. In short, the more accurate the prediction of the model, the lower the Brier score.

For comparison, we also tested a model in which predicted choice was set by a non-dynamic logistic function that mirrored the psychometric curve derived from the entire sessions' sample; we term this logistic function the no-history model. The difference in Brier scores is given in Fig. 4c, revealing that the history model provided a better prediction of the rats' choices across the range of $\Delta$speed values. The stronger improvement of the history model at the center than at the extremes of the stimulus range reflects the fact that the rats' classification of stimuli with $\Delta$speed close to 0 was more influenced by recent trials. The area under the receiver operating curve (AUROC) of the history model was 0.84 on average, while for the no-history model it was 0.83 (Fig. 4d, top), an improvement of around 0.5% ($p < 0.001$, corrected resampled $t$-test; Cohen's $d = 4.1$). As mentioned, Brier score improvement was higher when considering current stimuli in the center of the range, peaking at $\Delta \text{speed} = -1$ (likely due to a slight imbalance in the rats' decisions; see Fig. 4c and Supplementary Fig. S5b). Computing the AUROC for $\Delta \text{speed} = -1$ trials (Fig. 4d, bottom), the no-history and history models had values of 0.52 and 0.58, respectively (a 10% improvement; $p < 0.001$, corrected resampled $t$-test; Cohen's $d = 3.6$).

**Dynamics of the shift in decision criterion.** Previous sections showed that rats' choices can be predicted by a model where stimulus $n-1$ biases the trial $n$ choice by "pulling" the decision criterion. What are the dynamics of that attraction across the ITI? (See Fig. 1a for definition of ITI.) The decision criterion might be updated during stimulus $n-1$ presentation and then remain stable throughout the time span between successive trials, corresponding to the discrete, step-like updating of Eq. (3) and Fig. 4a. On the other hand, the decision criterion could be non-stationary over the course of the ITI. A time-dependent criterion shift can be envisaged as taking one of two opposing forms. First, the criterion might be instantaneously attracted to stimulus $n-1$, before relaxing toward a second attractor, perhaps related to the central tendency of past stimuli. This would result in stimulus $n-1$ imposing a bias that is initially strong but weakens as time passes. Alternatively, the criterion might be attracted to stimulus

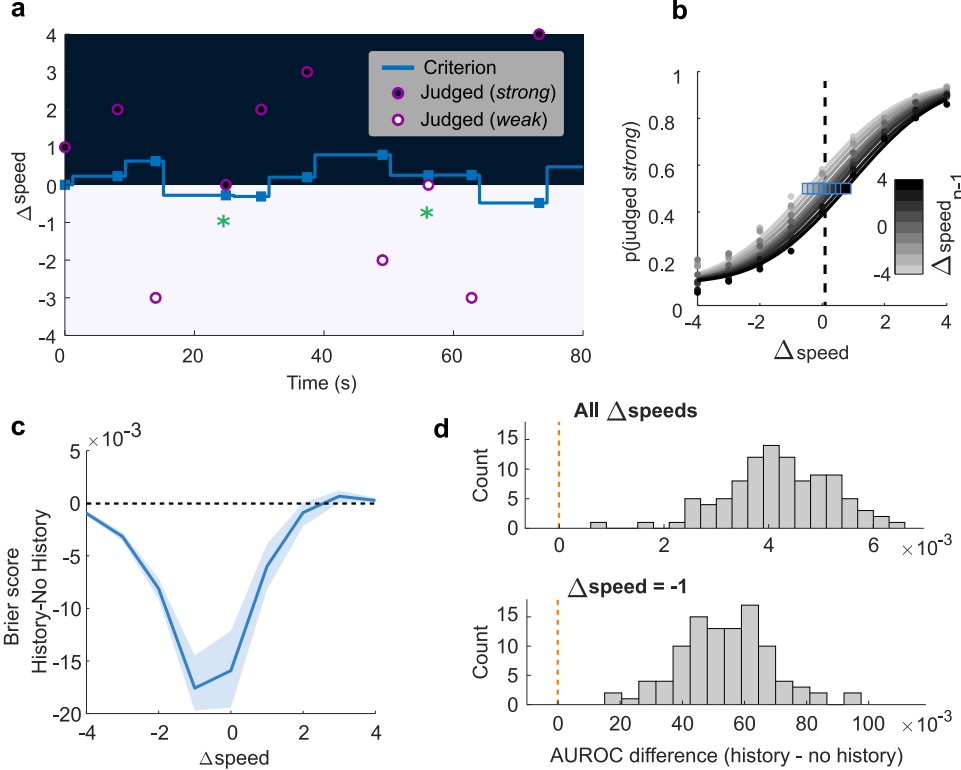

**Fig. 4 Discretized trial-by-trial model of criterion updating. a** Behavior of the model for a 10-trial sequence; see main text for definitions. **b** Effect of $\Delta speed_{n-1}$ on trial $n$ psychometric curves based on discretized history model. Horizontal shifts were set by $\mu_n$. Blue squares denote points of subjective equality (PSE) and the black dashed line illustrates PSE computed with all trials merged. **c** Prediction improvement (Brier score decrease) using the discretized history model with respect to the no-history model, for each of the nine stimuli in trial $n$. Blue line is the median and shaded area the interquartile range of the cross-validations. **d** Comparison of the area under the ROC curve (AUROC) of the discretized history model vs. the no-history model for different cross-validation test sets. The upper histogram shows the AUROC difference computed by considering predictions for all stimuli in trial $n$. The lower histogram shows the AUROC difference computed by considering predictions for $\Delta speed_n = -1$ (the stimulus value for which the model afforded the largest benefit). Note the different abscissae scales. Each AUROC value derives from one round of cross-validation and the histograms plot the outcome of 100 rounds. Source data are provided as a Source Data file.

$n-1$ progressively. This would result in stimulus $n-1$ imposing a bias that is initially weak but strengthens as time passes.

Because the behavioral task was self-paced by the rat, the distribution of ITIs was roughly normal but with a long tail (Fig. 5a). We excluded the longest 20% of ITIs from further analyses (gray bins), as these were associated with task-unrelated behaviors such as grooming or napping, and we divided the remaining ITIs into quartiles (Fig. 5a). Next, we assessed how stimulus $n-1$ modulated the trial $n$ psychometric function parameters in relation to the ITI from $n-1$ to $n$. The slope of the psychometric function was not significantly modulated by ITI (GLM interaction test between the effect of ITI and $n-1$ $\Delta$speed on slope, $p = 0.47$; ITI main effect $p = 0.18$). Further analyses (Supplementary Fig. S6) indicated a significant influence of ITI on psychometric function midpoint but not on lapse rates. According to this analysis, we modeled the effect of the ITI through the midpoint of the psychometric curves, setting lapse parameters to the average level of each rat. Figure 5b shows psychometric curves on trial $n$, averaged across six rats, separated according to $\Delta$speed of stimulus $n-1$ (−4 or 4) and further separated according to the ITI from $n-1$ to $n$ (shortest quartile in light blue, longest in dark blue). After a long ITI, $n-1$ exerted a stronger effect, signifying that the shift in decision criterion toward $\Delta$speed of stimulus $n-1$ did not occur in a single step.

Confirming the time-dependent shift in decision criterion, Fig. 5c reveals that the regression line, fitting the bias on trial $n$

imposed by stimulus $n-1$, was steeper after a long ITI (longest two quartiles combined) than after a short ITI (shortest two quartiles) (interaction term between previous $\Delta$speed and ITI durations, $p = 0.014$). We performed control analyses excluding potential confounds (see Supplementary Fig. S7). First, we replicated Fig. 5b, c after excluding previous incorrect trials (Supplementary Fig. S7a, b). Furthermore, using only the first 150 trials of each session (when motivation or urgency might be stronger), the ITI had the same effect on trial $n$ choice as for the full session. There was a difference of just 50 ms in median ITI for the first 150 trials of each session compared to the whole session (7.57 s versus 7.62 s), suggesting that the pace of the task was stable within sessions. The difficulty of trial $n-1$ (i.e., the closeness of $\Delta$speed to the category boundary) did not affect the ITI from $n-1$ to $n$ (Supplementary Fig. S7c). Finally, a GLM analysis (Supplementary Table T2) uncovered a significant repulsive effect of stimulus $n-2$, which was reduced with increasing ITI from stimulus $n-1$ to $n$. This is explained by the memory of $n-2$ being "written over" as $n-1$ exerts a progressively stronger effect. Accordingly, with reduced ITI from stimulus $n-1$ to $n$ the effect of $n-2$ was larger. Taken together, these tests argue against the possibility that slow fluctuations in ITI during the course of the session co-varied with (and might partially account for) the trial-by-trial stimulus history effect.

Human subjects received the analogous stimulus set on the right index finger and judged each vibration as "weak" or

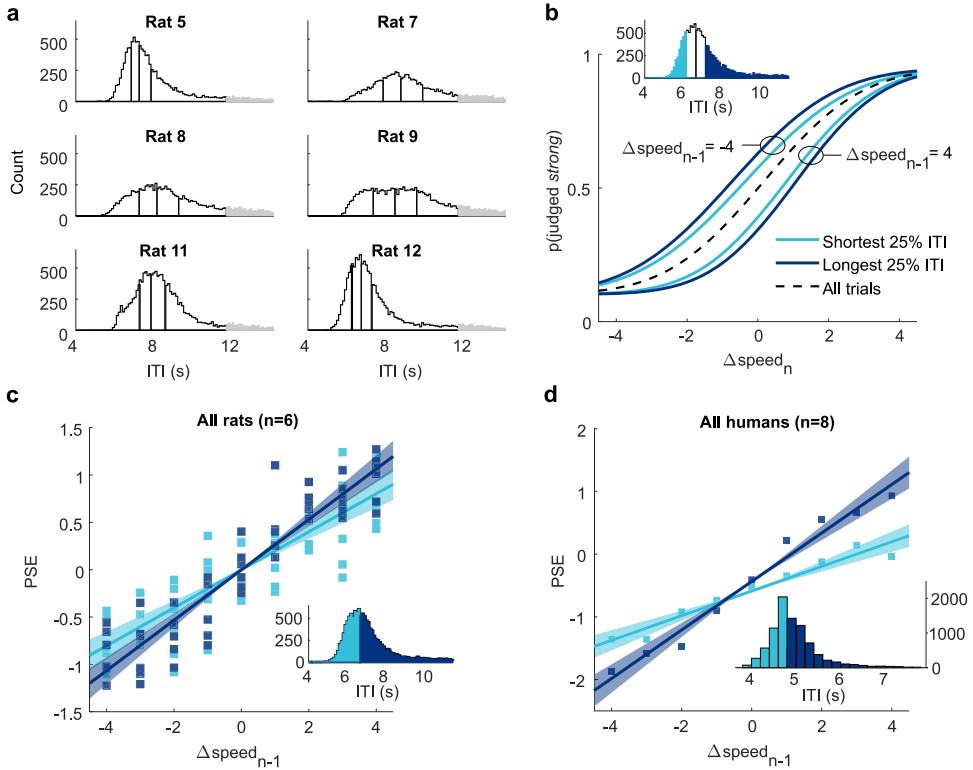

**Fig. 5 Stimulus-dependent bias builds up between trials. a** ITI histograms for each of the six rats, based on all sessions. ITIs were binned into four quartiles, after long outlier ITIs (gray) were excluded. **b** Psychometric curves on trial $n$ plotted according to $\Delta$speed of stimulus $n-1$ and ITI. Inset shows the shortest and longest quartiles of ITI for an example rat. Dashed line is the average psychometric curve on all trials. **c** Slope of the regression line fit between bias (PSE) and previous trial $\Delta$speed for all rats, separated by shortest two ITI quartiles (light blue) and longest two ITI quartiles (dark blue). Shading represents 95% confidence intervals. ITIs were split around each rat's median. **d** Slope of the regression line fit between bias (PSE) and previous trial $\Delta$speed for eight human subjects, separated by shortest two ITI quartiles (light blue) and longest two ITI quartiles (dark blue). Regression lines were fitted on data after 'short' and 'long' ITIs, after residualizing PSEs from the effect of previous choice. Only previous correct trials were considered. Shading represents 95% confidence intervals. ITIs were split around each human subject's median. Inset: ITIs were pooled across subjects to form a single histogram, and were then divided around the subjects' grand median. Source data are provided as a Source Data file.

"strong," paralleling the experiments in rats (see Supplementary Fig. S8). Stimulus $n-1$ exerted a repulsive effect on the trial $n$ choice—a low-speed vibration on trial $n-1$ led to increased likelihood of a "strong" choice on trial $n$, high-speed $n-1$ vibration led to "weak" choice on trial $n$—and the bias imposed by the preceding stimulus grew with increasing ITI, generalizing the key psychophysical findings from rats to humans (Fig. 5d). In contrast to rats, in humans we also found a prominent attractive choice effect (in line with previous findings[33,34]). For the purpose of isolating the stimulus history effect and its dynamics we factored out the choice effect (see Supplementary Fig. S9).

**Continuous model of history-dependent choices.** The time-dependent shift of the decision criterion toward $\Delta$speed$_{n-1}$ requires the model for discrete criterion updating (Eq. (3)) to be reformulated. The shifting decision criterion, $\mu(t)$, is now described by an exponential convergence toward the stimulus $n-1$ memory trace:

$$\mu(t) = \mu(t_0) \cdot e^{\left(-\frac{t}{\tau}\right)} + \Delta speed_{trace}(t) \cdot \left(1 - e^{\left(-\frac{t}{\tau}\right)}\right) \quad (5)$$

where $\mu(t_0)$ is the decision criterion at the presentation of stimulus $n-1$ (equivalent to $\mu_{n-1}$), $t$ is the elapsed time after presentation of stimulus $n-1$, and $\Delta$speed$_{trace}(t)$ is the stimulus $n-1$ memory trace. The convergence of $\Delta$speed$_{trace}(t)$ toward

$\mu(t)$ is expressed as follows:

$$\Delta speed_{trace}(t) = \Delta speed_{trace}(t_0) \cdot e^{\left(-\frac{t}{\tau}\right)} + \mu(t) \cdot \left(1 - e^{\left(-\frac{t}{\tau}\right)}\right)$$

$$(6)$$

where $\Delta speed_{trace}(t_0) = \Delta speed_{n-1}$. The model posits that stimulus $n-1$ instates a neuronal representation that, once used for the trial $n-1$ decision, persists thereafter as a labile memory trace, a short-term stimulus buffer resembling that demonstrated in earlier work[35,36]. By setting $t$ to the ITI between $n-1$ and $n$, Eqs. (5) and (6) give the estimated decision criterion on trial $n$. Reworking these two equations, or using symmetry considerations, one can see that when $t$ goes to infinity, $\mu(t) = (\mu(t_0) + \Delta speed(t_0))/2$; that is, the decision criterion would asymptotically reach a value midway between its previous value, $\mu(t_0)$, and the $n-1$ stimulus. While the parameter $\tau$ in the initial model referred to a decay in the weight of past stimuli, discretized as trial numbers, in the revised model $\tau$ is a time-continuous parameter that dictates how rapidly the decision criterion is drawn to the memory trace of the most recent stimulus.

The time constant, $\tau$, is the only free parameter in the model of Eqs. (5) and (6); it is fitted by maximizing the model's performance in predicting the rat's actual choices (see Methods). Figure 6a shows the trajectory of the computed decision criterion for the same stimulus sequence to which the discretized model was applied previously (the discretized model criterion is carried over from Fig. 4a as light gray dotted line). As before,

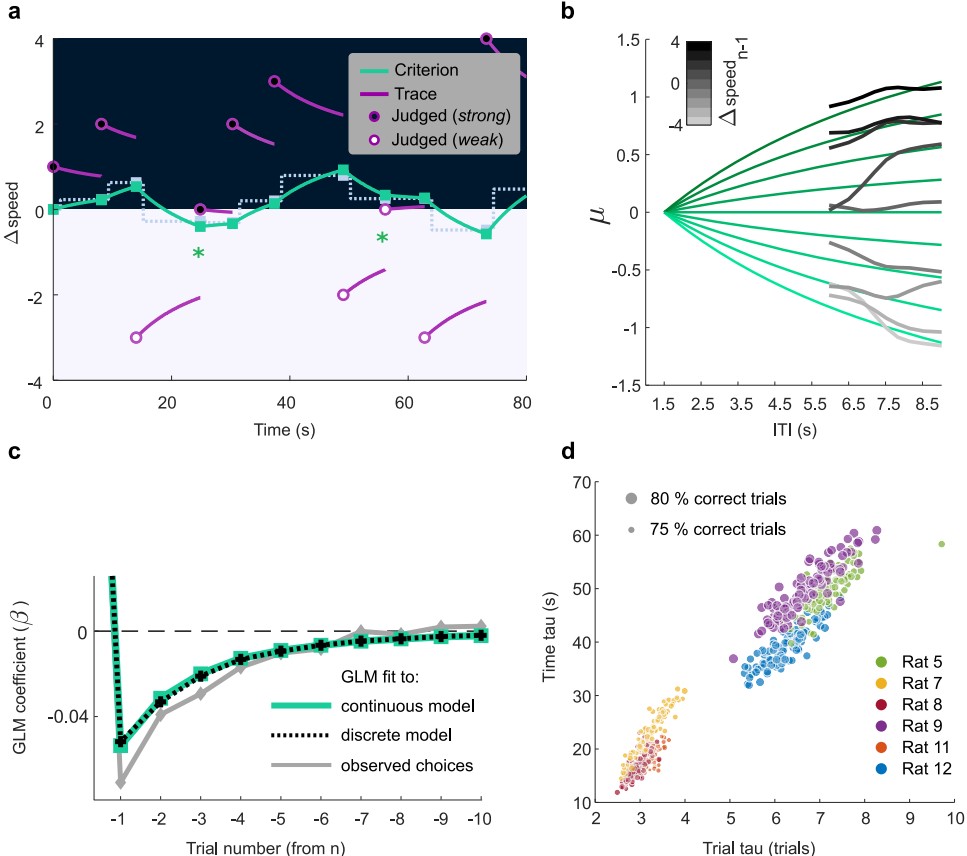

**Fig. 6 Continuous model of criterion updating. a** Behavior of the model for the same 10-trial sequence illustrated in Fig. 4a. Green trajectories are the model's computed decision criterion, with $\tau = 30$ s. For comparison with the continuous model, the discrete model computation is shown by the dotted line. **b** Gray traces give the observed trial $n$ psychometric function midpoints, in the manner of Fig. 3a, as the ITI from $n-1$ to $n$ grows, smoothed with a moving average window of 8 bins. Traces are sorted according to $\Delta$speed of stimulus $n-1$ (see gray scale). Green traces give the model's predicted trial $n$ psychometric function midpoints, as the ITI from $n-1$ to $n$ grows. **c** Comparative performance of the continuous and discretized history models. The same GLM of Fig. 3f (here shown in gray) was fitted to artificial choices simulated from trial to trial with the discrete and the continuous model. **d** Optimized time constants, $\tau$, are estimated in each cross-validation partition and are shown for both discrete and continuous models in six rats. The size of the points corresponds to the average performance of the rat in the selected subset of sessions. Source data are provided as a Source Data file.

circles denote presented $\Delta$speed values. The black area covers $\Delta$speed values rewarded as "strong" while the white area covers $\Delta$speed values rewarded as "weak." On trials denoted by filled and unfilled circles, the rat judged the stimulus as "strong" and "weak," respectively. The green trajectories depict the decision criterion generated by the model, and the green squares highlight the criterion value at the moment of stimulus presentation. The purple trajectories depict the memory trace of the most recent stimulus, $\Delta$speed$_{trace}(t)$, which approaches the decision criterion in a symmetrical manner. Correct predictions on the two trials with $\Delta$speed $= 0$ are denoted by the green asterisks.

Figure 6b illustrates how the model captures the "pull" exercised by stimulus $n-1$ on the next trial's psychometric function. The gray lines, sorted according to $\Delta$speed of trial $n-1$, show the mean trial $n$ curve midpoints, $\mu_n$, in real data, as a function of increasing ITI, generated by pooling all trials from 329 sessions of six rats. The overlying green lines give the model's computed decision criterion over time, also sorted according to $\Delta$speed of trial $n-1$.

By formalizing the mutual attraction between the decision criterion and the most recent memory trace, the model offers plausible explanations for our previous observations. The trial $n$ psychometric curve midpoint approaches but never reaches $\Delta$speed$_{n-1}$ (Figs. 3a and 5b) because the $n-1$ memory trace is,

at the same time, attracted to and approaching the decision criterion.

If stimulus $n-1$ attracts the criterion that will be applied on trial $n$, how do stimuli even earlier than $n-1$ exert their effects? A memory for past stimuli is built into the model, inasmuch as the criterion upon which stimulus $n-1$ acts had been previously influenced by $n-2$, and so on. In other words, stimulus history is continually built up as new trials are sequentially embedded within the running value of the criterion.

As with the discretized model, we estimated the probability of the choice "strong" on trial $n$. We inserted the criterion, $\mu(t)$, estimated by Eq. (5), as the midpoint parameter of the logistic function (see Eq. (4)), and then assessed the distance from the model-based psychometric curve to the rat's choice. We performed a cross-validation by repeatedly partitioning the sessions into two sets, one used for estimating $\tau$ (training; 80% of sessions) and one for testing (20% of sessions), and compared the continuous model's predictions to the discretized model's predictions for the same data.

When considering all trials in the data set, AUROC was slightly but significantly higher for the continuous model compared to the discrete model (0.015% higher; $p < 0.001$, corrected resampled $t$-test; Cohen's $d = 0.66$; see Supplementary Fig. S10a). When considering exclusively the judgments for $\Delta$speed $= -1$, for which decisions were the most history-driven (likely due to a

slight imbalance in the rats' decisions, see Fig. 4c and Supplementary Figs. S5b and S6c), the increase in AUROC was greater (about 0.32%; $p < 0.001$, corrected resampled $t$-test; Cohen's $d = 0.8$; see Supplementary Fig. S10b). Brier score was also significantly lower (corrected resampled $t$-test, $p < 0.001$). Together, these results indicate a higher predictive power of the continuous model with respect to the discrete model.

To investigate how the trials prior to $n - 1$ would be expected to influence trial $n$ choices according to the two models, we allowed both the discrete and continuous models, with $\tau$ already fitted, to predict rats' choices for the entire set of trials in their real temporal order. From the resulting series of choices, we used the GLM first employed for Fig. 3f to solve for the weighting of the $\beta$ coefficients (Eq. 2). In Fig. 6c, it can be seen that for both models the coefficients decay with an exponential trend. Because the analysis pools trials with different preceding ITIs, the continuous and discrete models converge. For comparison, the GLM coefficient weights fitted on the observed rat data are shown in gray, carried over from Fig. 3f.

The values of $\tau$ derived from the discretized and continuous models shed light on the strategies employed by the rats. In Fig. 6d, the data set from each rat was subsampled in partitions of 80% of sessions and for each partition the optimal $\tau$ was computed. For the discretized model, rats' choices were best fit with $\tau$ in the range of 2.5–7 trials. For the same sessions, $\tau$ derived from the continuous model was highly correlated ($R = 0.92$, $p < 0.001$), ranging from 10 to 50 s. Thus, both models capture the rats' relative weighting of past sensory experience in the current choice. The better predictive power of the continuous model originates in its ability to factor in the ITI, the time period across which the most recent stimulus has attracted the decision criterion.

Under the conditions of this study, the ideal history-dependent decision criterion would incorporate, with equal weight, all past trials. Rats did not do so (nor did humans), but performance across rats (size of points in Fig. 6d) uncovered two clusters of three rats each—those with shorter $\tau$ were more biased by the most recent trial and performed more poorly while those with longer $\tau$ were closer to ideal observers and performed better. To further investigate the functional significance of $\tau$, we bootstrapped the rats' sessions, fitting $\tau$ and estimating sensitivity ($\sigma$, the standard deviation of the underlying cumulative Gaussian distribution) for each sample. This revealed an overall within-rat correlation between the time constant and the sensitivity ($R = -0.1$, $p = 0.015$; negative correlation means that for longer $\tau$ there was a higher sensitivity). The fact that rats' $\tau$ magnitudes correlated with their performances, while being uncorrelated with the rats' average ITI durations ($R = -0.03$, $p = 0.42$), supports the view of $\tau$ corresponding to a relevant behavioral measure, explaining a significant degree of accuracy.

## Discussion

**Sorting out the effects of recent trials**. Although it can be appealing to conceive of sensory-perceptual systems as ideal observers that reliably map physical inputs onto the appropriate responses, experimental data frequently reveal variability in perceiving or acting upon separate presentations of an identical stimulus. Performance of a perceptual task appears to involve less a rigid stimulus-to-response transformation than a flexible adjustment to the full experimental context, including the history of rewards, choices, and stimuli[6–8,12–14,16,18,19,34,37–52]. By sorting the sequences of trials according to each of these factors, it becomes possible to disentangle their contributions.

While at early stages of skill learning and rule learning, choices are driven by reward contingencies[41–43], in tasks that promote faster decision making, such as when stimulus presentation is self-terminated by the subject, rewarded choices can attract those in subsequent trials, known as "win-stay-lose-switch" (e.g.,[44–46,53]). Attractive serial effects also characterize perceptual judgments based on categorical or object-related sensory attributes such as orientation or face identity, as if the category itself were stabilized[7,8,12,16,17,47–49]. By contrast, studies in which subjects judge uncertain stimuli distributed in a continuous manner along a physical gradient[8,12] frequently highlight a repulsive perceptual effect (once the preceding choices per se are factored out) in that the current stimulus is felt to be more distant from that of the preceding trial than would be expected if history were not considered.

In the present task, individual vibrations delivered to the rats' whiskers had to be categorized according to $\Delta$speed as "weak" or "strong." Since the stimuli were distributed along a continuum, with no qualitatively distinct attractors or qualia, rats were required to apply an internal threshold, or decision criterion, to each instance of the stimulus. Under these conditions, a robust repulsive stimulus history effect prevailed: after a high-speed stimulus, the next stimulus was judged as weaker than it would otherwise, and after a low-speed stimulus, the next stimulus was judged stronger (Fig. 3a).

Trial-by-trial analysis revealed the effects of previous choice (Fig. 3b) and reward (Fig. 3c) to be minor, leaving the physical magnitude of the past stimuli as the major factor responsible for variability in perceptual judgments in the given task.

**Dynamics of the shift of the decision criterion**. The robustness of the repulsive stimulus history bias allowed us to focus on its dynamics across different time scales. While choices at the outset of each test session were unaffected by the previous session (Fig. 2c), within a session the stimulus history effect was retained over several trials (Fig. 3d). At a finer timescale, the repulsive bias grew with inter-trial duration (Fig. 5b, c), contrary to the intuition that stimulus $n - 1$ might exert an immediate but transitory effect on trial $n$. This unexpected growth led us to hypothesize that the gradual attraction of the decision criterion, $\mu_n$, to the stimulus of trial $n - 1$, might be the root cause of the repulsive stimulus history effect. We tested the hypothesis using a model where a time constant $\tau$, the only free parameter, quantifies the attraction of the decision criterion toward $\Delta$speed$_{n-1}$ (Eqs. 5 and 6). With the decision criterion now defined as a time-dependent variable, $\mu(t)$, choices could be modeled by centering a sigmoid-shaped response probability function on the criterion. The proposed model significantly improved predictions of the rats' actual choices, as compared to a response probability function that was not shifted according to stimulus history (Figs. 4c, d and 6c).

The choice boundary in our study resembles the quantity referred to in the literature by the term decision criterion[12,30] or as implicit standard[13]. Its calculation as the center of the weighted distribution of past stimuli resembles the quantity sometimes termed the prior distribution[36,50,54–56].

In earlier work, it was unknown what mechanism might determine the weighting of past stimuli, especially in specifying effects as deriving from elapsed time versus the number of successive stimulus presentations[17,51,52]. By gathering a broad range of ITIs, with each interval densely sampled, we were able to dissociate the two potential mechanisms underlying the repulsive stimulus history effect. In our formulation, only the most recent trial acts to update the decision criterion. Each successive stimulus is thus embedded within the criterion by virtue of its attractive pull, and the criterion is handed over for the execution of the next trial, after which it is updated again. No terms are

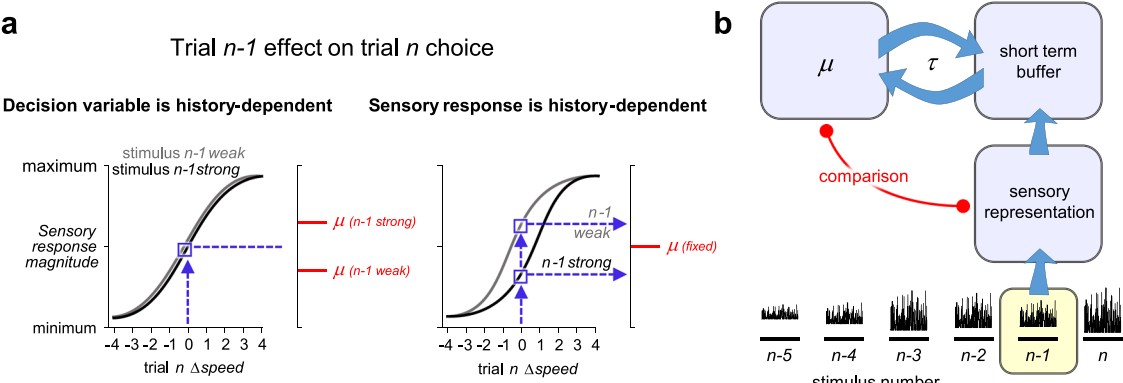

**Fig. 7 Candidate neuronal mechanisms underlying trial history effects. a** Judgment of $\Delta$speed $= 0$ on trial $n$ may depend on stimulus $n - 1$ in two ways. Left: sensory response on trial $n$ is invariant to stimulus $n - 1$. The criterion $\mu$ (red bar) that determines the subject's categorization of the sensory response is history-dependent. Right: sensory response on trial $n$ varies according to stimulus $n - 1$. The criterion $\mu$ (red bar) that determines the categorization of the sensory response is fixed. **b** In this model, the $n - 1$ sensory representation is compared (red line) to the decision criterion, $\mu$, and the comparison outcome is converted to choice. During stimulus presentation, $n - 1$ is also loaded into a short-term memory buffer, which successively attracts $\mu$ with a time constant, $\tau$. When stimulus $n$ is presented after an inter-trial interval, its sensory representation is compared to the new criterion, which has now been updated through its attraction to $n - 1$.

required in our model to specify the weights of more distant past trials; rather than a multifactorial generalized linear model[34,37,44,50], our model is reduced to one variable that quantifies how rapidly the most recent stimulus pulls the criterion.

**Perceptual bias versus decisional bias.** Our model posits that the stimulus $n$ sensory response is projected onto the decision variable, which converts its sensory input into a choice according to the current decision criterion, $\mu_n$ (Figs. 4a and 6a). The mapping function from $\Delta$speed$_n$ onto the decision variable is modulated by preceding stimuli. Two ways in which the brain may mediate the stimulus-to-decision mapping[57] are illustrated in Fig. 7a. In the left panel, the stimulus presented on trial $n$ (INPUT) evokes a sensory response (OUTPUT) that is invariant to $\Delta$speed$_{n-1}$: gray ($n - 1$ weak) and black ($n - 1$ strong) curves are overlapping. Suppose that on trial $n$, $\Delta$speed $= 0$. The corresponding sensory response (blue square) is projected horizontally onto the axis of the decision variable. The criterion (red line) determines the categorization of the sensory input; inputs above and below the criterion are judged as strong and weak, respectively. The criterion itself is history-dependent, moving downwards after a weak $n - 1$ and upwards after a strong $n - 1$, echoing the plots of Figs. 4a and 6a. These criterion shifts lead to opposite choices for two instances of $\Delta$speed$_n = 0$.

An alternative form of stimulus-to-decision mapping is illustrated in the right panel of Fig. 7a. Here, stimulus $n$ evokes a sensory response that varies according to stimulus $n - 1$: gray and black curves are separated. Two different, history-dependent outputs for $\Delta$speed$_n = 0$ (blue squares) are projected onto the decision variable. The criterion (red bar) is fixed; it is the shifting sensory representation that leads to opposite choices for two instances of $\Delta$speed$_n = 0$. The second form of shift calls to mind the rescaling of the sensory system coding metric according to ongoing context, a kind of adaptation believed to maximize information transmission in a varying environment[58,59], where some effects have been shown to decay exponentially over time[60,61].

The two candidate mechanisms of Fig. 7a are both compatible with our computational model—in Eqs. (3) and (5), $\mu_n$ may be taken as the decision criterion or, equivalently, as the input/output curve midpoint. However, they lead to very different predictions. According to the framework with history-dependent

decision variable (left panel), trial-to-trial sensory responses must be encoded stably in somatosensory cortex; in later stages of processing, perhaps corresponding to the frontal cortex targets of somatosensory cortex, neuronal populations must trigger opposite actions for the same input, in a history-dependent manner. By contrast, according to the framework with history-dependent sensory response (right panel), the encoding of stimuli at an early processing stage will already be influenced by stimulus history. In later stages of processing, the choice to be triggered is a stable function of that history-dependent sensory input.

**Functional significance of the time constant $\tau$.** In both frameworks, the history-dependent feature, be it at the decision making stage or at the sensory coding stage, must adapt to the most recent stimulus by dynamics defined by the time constant $\tau$, a quantity that is independently observable by behavioral measures. The values of $\tau$ (whether computed by the continuous model or the highly correlated discretized model) evidence two clusters of rats (Fig. 6d). Three rats had short time constants, and therefore expressed high history-dependent choice volatility ("hot rats") and the remaining three had longer time constants and correspondingly lower volatility ("cold rats"). Strikingly, the two clusters are distinguished by their average performance: cold rats performed better than hot rats. It must be kept in mind, however, that the long $\tau$ that is advantageous for the fixed boundary condition of the present task (approaching infinity in the ideal case) is not necessarily ideal for a volatile stimulus context, for example if the stimulus distribution and reward boundary were programmed to move every ten trials. In the latter case, a decision boundary built on the history of only recent stimuli may be normative if one assumes that perceptual decision-making systems have selected for mechanisms that allow them to rapidly adjust to volatile environments, where stimulus range and categorical boundary may shift unexpectedly. Adaptation to this sort of environment could be implemented in the brain by hierarchical Bayesian mechanisms[62,63]. Is $\tau$ fixed for a given rat, or is it recalibrated according to the statistics of the world? In the latter case, the long-$\tau$ rats in the present experiment would compress their integration time in a more volatile world; if $\tau$ is a fixed, innate characteristic of the rat, then the short-$\tau$ rats in the present experiment would have an advantage in a more volatile world. The finding that the volatility of an individual rat's criterion varied from session to session (and generated corresponding

changes in discrimination performance) suggests that $\tau$ is not rigid.

**Model of two interacting modules**. The hypothesis of Fig. 7a, left panel, can be further envisioned as a set of interacting modules (Fig. 7b). To execute trial $n-1$, the corresponding sensory representation is compared (red line) to the decision criterion $\mu$. As stimulus $n-1$ is encoded, perceived, and acted on, it is simultaneously loaded into a short-term buffer. Across the ITI, the buffer attracts the criterion by a time constant, $\tau$ (Eq. (5)). The next stimulus, $n$, is again compared to the decision criterion, which has now been updated through its interaction with the short-term buffer.

The same model may be generalized to account for the effect of stimulus history in a different perceptual memory behavior. In a delayed comparison working memory task[26,64], the comparison stimulus ($n$ in our terminology) is measured against the memory of the base stimulus ($n-1$ in our terminology) which is held in the short-term buffer[35,36,65]. Memory of the base stimulus is known to be attracted toward a prior related to preceding stimuli, a phenomenon known as "contraction bias"[6,66]. Contraction bias might be explained if $\mu$ exerts an attractive force on the short-term buffer, reciprocating the attraction exerted by the short-term buffer on $\mu$ (opposing arrows in Fig. 7b). Consistent with this supposition, the prior can be manipulated separately from the short-term buffer[50]. The short-term buffer is not purposefully engaged in the present reference memory task; nevertheless, it plays a role by attracting $\mu$ (see also[67]). Complementarily, the prior ($\mu$) is not purposefully engaged in working memory tasks; nevertheless, it plays a role by attracting the short-term buffer. The current form of the model considers the strength of attraction between the decision criterion, $\mu(t)$, and the stimulus $n-1$ memory trace to be equal, causing the short-term and long-term buffers to converge symmetrically. Future studies will explore the possibility that the attraction is asymmetric; indeed, the interaction between $\mu(t)$, and the stimulus $n-1$ memory trace might even be task-dependent.

Such interrelations, if further evidenced in a more complete set of human psychophysical behavioral experiments, could have relevance to clinical studies[68].

## Methods

All protocols conformed to international norms and were approved by the Ethics Committee of SISSA and by the Italian Health Ministry (license numbers 569/2015-PR and 570/2015-PR).

**Subjects**. Eight male Wistar rats (Harlan Laboratories, San Pietro Al Natisone) were trained/handled on a daily basis and caged in pairs. They were regularly checked for their health and welfare conditions and provided with daily environmental and social enrichment. They were maintained on a 12/12-h light/dark cycle. To promote motivation in the behavioral task, rats were water-restricted between daily testing sessions, while continuously having free access to food in the cage. They were tested on each working day in sessions of about 1 h. Water was freely available on days not followed by experimental sessions.

Eight human subjects were recruited among university students or employees through a university mailing list. In order to participate in the experiment, human subjects received detailed instructions and gave informed consent in written form.

**Behavioral task**. An LED placed on the nose poke signaled to the rat that a trial could be initiated by crossing the optical sensor inside the nose poke. In the nose-poking position, the rat's right whiskers touched the plate (Fig. 1a). Triggering the nose poke optical sensor led to a 400 ms delay followed by a 500 ms vibration of the plate. The vibration was followed by a random delay (400–600 ms), after which an auditory go cue instructed the rat to withdraw and choose one of the two spouts. Infrared beams at the entryway to each spout detected the rat's decision. According to vibration speed one of the two spouts was enabled to deliver fluid reward, e.g., left spout for vibration speeds above the category boundary and right spout for vibration speeds above the category boundary. No reward was delivered after incorrect choices. After presentation of the stimulus on the category boundary, reward was assigned randomly to one of the two spouts. After incorrect choices, the

nose poke sensor was inactivated for 1500–3500 ms, forcing the rat to wait for the next trial. ITIs were measured from one crossing of the nose poke optical sensor to the next one (Fig. 1a). Typically, the training or testing session lasted about 1 h and included ~300 trials. When a rat obtained a consistent threshold of performance, defined as >75% correct for five consecutive sessions, the training phase was considered to be completed and the remaining sessions were taken as the data for analysis.

Human subjects performed a modified version of the same task. After receiving the stimulus and the go cue, they had to report their choice by pressing one of two buttons. Buttons were asymmetrically placed in order to minimize motor biases (see also Supplementary Fig. S8). Subjects received feedback (correct/incorrect) on each trial through a computer monitor and headphones. The final amount of money given to each subject was proportional to the percentage of correct trials in the task, and subjects were informed about this aspect prior to the experiment. Each subject performed two sessions of 750 trials each. Each session was divided into three blocks of 250 trials each, separated by 5-min breaks.

**Vibrotactile stimuli**. For rats, the stimulation medium consisted of a rectangular plate ($20 \times 30$ mm) connected to a motorized shaker (Bruel and Kjar, type 4808) to which velocity values were sent as analog signals, moving the plate along the rostro-caudal axis. Stimuli were vibrations of the plate made of low-pass filtered white noise, obtained as in[26]. Velocity values of one stimulus were sampled at 10 kHz from a normal probability distribution function with 0 mean. There were 50 seeds (specific time series) available for each velocity standard deviation. The noise was low-passed through a Butterworth filter with 150 Hz cutoff, amplified, and sent as voltage input to the shaker motor. Vibration speed was quantified as the mean absolute value of velocity (i.e., the mean speed, $sp$), equal to the normal distribution's standard deviation multiplied by $\sqrt{(2/\pi)}$. This physical feature has been identified as allowing high acuity discrimination of whisker-mediated vibrations in rats[69]. In every session, there were nine linearly spaced values of $sp$. Unless stated differently, as in the data of Fig. 2, on each trial the software randomly selected one of these nine mean speeds with a probability of 0.11. In order to discourage the rat from forming the habit of choosing one side repetitively, not more than three sequential trials of one category were presented.

For humans, the stimulus was delivered by means of a rounded probe that vibrated along the axis of the rod, and was controlled by the same model of shaker motor used in rats (see also[27]). Stimuli delivered to human subjects on the fingertip were the same as those used in rats except that the ranges of $sp$ values were chosen among four different difficulties, depending on the performance on a short set of 75 preparatory trials.

**Experimental apparatus**. The apparatus, custom-built by CyNexo (https://www.cynexo.com/), consisted of a Plexiglas box measuring $25 \times 25 \times 38$ cm (height × width × length) that was located inside a sound-proof and lightproof chamber. Reward spouts on each of the two sides of the apparatus, fashioned from metal tubes with a plastic lip, delivered 0.03 mL of juice/water. They were actuated by syringes controlled by pressure-pumps, on correct choices. A rounded head-port on the front wall of the apparatus allowed rats to access the nose poke, a circular aperture of 0.85 cm diameter. During sessions the box was closed with a Plexiglas cover and monitored with a camera placed on top of the apparatus. All the software for the control of the rat and human experiments were written in-house in LabVIEW (National Instruments, Austin, TX).

**Psychometric curve estimation**. All analyses were performed in MATLAB (MathWorks, Natick MA). To estimate psychometric functions, we computed for each stimulus vibration $sp$, the proportion of trials in which the rat responded by going to the side corresponding to the "strong" category. We fitted the data with a probit function including asymmetric lapse parameters:

$$p(\text{strong}) = \gamma + (1 - \gamma - \lambda) \cdot \frac{1}{2}\left[1 + erf\left(\frac{\Delta \text{speed} - \mu}{\sigma\sqrt{2}}\right)\right] \quad (7)$$

where $\Delta$speed is a $sp$ value mapped to the scale of $-4$ to $4$ (see Results), $\mu$ is the midpoint parameter, $\sigma$ the slope parameter, $\gamma$ and $\lambda$ the lower and higher asymptotes of the function, respectively, corresponding to lapse rates (see[70]). Parameter values were estimated by maximum-likelihood using the MATLAB function fmincon. For GLM analyses we employed MATLAB's fitglm function.

**Fitting and cross-validation of recursive models**. To fit recursive models to the rats' choices as a function of previous stimuli, first, we randomly selected 80% of the behavioral sessions in the data set. Within each session selected, we recursively predicted the probability of the rat's "strong" judgment for each trial, depending on previous stimuli and ITIs. For fitting and validating the history-dependent logistic function, we employed the Brier score error term, defined as follows:

$$BS = \frac{1}{N}\sum_{n=1}^{N}(p(\text{strong})_n - \text{choice}_n)^2 \quad (8)$$

where $N$ is the total number of trials, $p(\text{strong})_n$ is the probability of the rat giving the "strong" response in a specific trial $n$ according to the model's psychometric

function on that trial and choice$_n$ is the observed outcome for that same trial (a binary variable equal to either 0 or 1).

To estimate the time constant $\tau$, we approximated the minimum of the Brier score function by means of a customized simulated annealing algorithm. The remaining 20% of the behavioral sessions of the same rat served as test sessions for the model with the $\tau$ estimated in the training set. For each rat, we repeated training and testing 100 times, by selecting different random session samples.

**Reporting summary**. Further information on research design is available in the Nature Research Reporting Summary linked to this article.

## Data availability

The following data employed in the current study have been deposited in a public repository (https://osf.io/hux4n): rat behavioral data used for the analyses of the trial-by-trial history effects and for fitting and evaluating the recursive models of stimulus history. Human behavioral data used for the analyses of trial-by-trial history effects. Any additional information will be available from the authors upon reasonable request. Source Data are provided with this paper.

## Code availability

The algorithms employed for predicting behavioral responses according to the discrete and continuous models of stimulus history (MATLAB), as well as codes used to produce the main plots analyzing the impact of different history variables on current decision, are available on OSF (https://osf.io/xkmy5). Any additional information will be available from the authors upon reasonable request.

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

## Acknowledgements

We acknowledge the financial support of the Human Frontier Science Program (http://www.hfsp.org; project RGP0015/2013), European Research Council advanced grant CON-CEPT (http://erc.europa.eu; project 294498), Beneficentia Stiftung, and Fondo Sociale Europeo – Programma Operativo 2014/2020. The Regional Laboratory for Advanced Mechatronics, LAMA FVG (http://lamafvg.it) supported the design and construction of custom instrumentation. We thank Arash Fassihi, Nader Nikbakht, Francesca Pulecchi, and Jacopo Rigosa for helpful discussions and valuable insights and Marco Gigante and Fabrizio Manzino (CyNexo) for their invaluable technical assistance. Sara Mohammed and Mauro Dall'Argine assisted in animal training and behavioral data acquisition.

## Author contributions

I.H., S.R. and M.E.D. designed the rat and human experiments with the assistance of A.S.; I.H. and A.S. carried out the rat and human data collection, supported by S.R.; I.H. wrote the analysis scripts, analyzed the data, and evaluated the models with close advisement of S.R. and occasional inputs from R.B., S.R. and I.H. devised and implemented the trial-by-trial recursive model. S.R. devised and implemented the model of two interacting modules with the help of R.B. and the consultation of M.E.D. and I.H.; I.H., S.R., R.B., and M.E.D. interpreted and discussed the results with contributions from A.S.; I.H., S.R. and M.E.D. wrote the paper, with inputs from the rest of the authors; M.E.D. performed the acquisition of the funding.

## Competing interests

The authors declare no competing interests.
