## [Peer Review File · Nature Communications]

Reviewers' Comments:

Reviewer #1:

Remarks to the Author:

Reinartz et al consider the fundamental question of how perceptual decision making depends on the prior history of stimuli, rewards and choices. This is a topic of considerable current interest and this study advances our understanding by applying quantitative analysis to data from rats and humans trained to discriminate the speed of a tactile vibration. The main claims of the study are that the primary history effect is a stimulus repulsion effect from the previous trial, that both humans and rats show the repulsion and that the data can be accounted for by a remarkably simple model.

Overall, I find the technical quality of the work to be high and the claims to be convincing. An important strength is that the authors have designed a task that delivers 1000s of trials; fundamentally it is this large data volume that permits them to tease apart effects of variable type, time course etc that previous studies knew to be potentially important but lacked the statistical power to resolve. The field needs these quantitative insights. Another strength is that the models used are simple, with very few parameters. This is crucial since it means that the models do not just replicate the data but add insight into what the key parameters that drive performance. A final strength – which separates this study from most others – is the evidence presented (from the human work) that the basic finding here is not merely a quirk of the rat (or indeed of the human) but a general principle of perception. Overall, the paper is likely to have significant influence on the broad community of researchers interested in perceptual decision making and its neural basis.

Medium:

1. There is much current interest in the idea that normative (especially Bayesian) theories are useful for understanding perception. On the face of it, perceptual repulsion (and the models proposed here, eg equation 3) are suboptimal given the task, and therefore present a challenge to normative theories. It would be useful to discuss this: do the authors think that Bayesian theory is challenged by their findings?

2. I buy that, in their task, perceptual repulsion is the major history effect. However, in Discussion, the authors state the effect of choice and reward are “negligible”. I am not fully convinced by this. First, there is some systematic separation between the blue and red curves in Fig 3b, indicating some modulation by n-1 choice. Second, there is also systematic separation between the green and purple curves in Fig 3c, indicating some modulation by reward. I do not dispute that these effects are most likely minor, but given that there does seem to be some effect, it would be interesting to take advantage of the authors’ modelling framework to quantify them. In particular, the model of Equation (3) predicts rat choice on 78% of trials. It would seem that the model (and/or their GLM) could straightforwardly be generalised to include choice and/or reward terms. This would allow numbers to be attached to the benefit (or not) of including choice/reward.

Minor:

In a few places, the reasoning is hard to follow. Please clarify lines 258-9 and, particularly, 271-8. In line 203-4, it is not clear why the result of Fig 2e cannot be due to a change in rule boundary. Line 131 – what is the basis for this assertion?

Line 170. The data points of Fig 2c are quite noisy (necessarily, since based on few trials) – can the authors exclude the null hypothesis that the separation shown could arise from chance?

Line 41. Other work, not cited, is relevant (Campagner et al, 2019; J Neurosci) since concerns rodent tactile decision-making, finding choice repetition effect.

Reviewer #2:

Remarks to the Author:

The study investigates the temporal dynamics of history biases in perceptual decision-making in rats and humans. The authors find that the recent stimulus history exerts a repulsive bias on current perceptual decisions, akin to well-known repulsive adaptation biases. Surprisingly, the authors find that the influence of the previous stimulus increases as the time between previous and current stimulus increases. Intuitively, I would have predicted to see the opposite pattern, and to my knowledge such an increase in history biases with increasing inter-trial intervals (ITIs) has not been reported before. The authors propose a new model in which the decision criterion is

gradually updated by each new stimulus. The manuscript is well written and the experiments appear thorough. In my opinion, the study presents an important addition to a rapidly increasing body of literature on history biases in perceptual decision-making. While I don't see any grave issues with the study, I have a few comments and suggestions that would need to be addressed before recommending the study for publication.

1. The authors show an intriguing pattern of increasing repulsion from the 1-back stimulus as the ITI between current and previous stimulus lengthens. The authors also find that 2- to 5-back stimuli still exert repulsive biases. I was wondering whether the authors have investigated whether those biases were also modulated by the current ITI. If I am not mistaken, their continuous model would only predict (/capture) modulations of the 1-back repulsion. I am asking this, because it is not entirely clear to me whether the variability in the ITI would only affect the integration of 1-back stimulus information with the decision boundary, or may also affect how the current stimulus is processed (e.g. rats/humans might be less alert following a long ITI; or ITIs might be longer in periods of inattention; or ITIs might increase over the course of the session and thereby correlate with attention) thereby potentially boosting the influence of multiple non-stimulus factors. I would suggest that the authors could run a GLM analysis similar to the one shown in Figure 3f, separately for current short and long ITIs, or add the current ITI as an interaction with previous stimulus evidence.

It would be also informative to test whether lapse rates or slope of the psychometric functions correlate with ITI, to rule out that effects of ITI reflect changes in current stimulus processing, rather than changes in the integration of decision criterion and 1-back stimulus information.

2. The authors use cross-validation to show that their continuous model outperforms the discretized and "no history" models. However, I think that the authors could do a better job in showing that their model actually provides an adequate fit to the empirical data. For instance, the continuous model predicts that as the ITI goes to infinity, the decision criterion should converge to a value halfway between the initial criterion and the previous speed. In Figure 5e, it looks like the effect of the previous stimulus plateaus rather quickly after 7 or 8 seconds. Does this reflect the asymptote which is halfway between initial criterion and the previous speed? In Figure 6b, which shows the model's prediction together with the data, the x-axis is cut off at 7.5 s or so (i.e. just around the beginning of the plateau). How does the fit look when extending the x-axis to 10 seconds (like in Figure 5e)?

Relatedly, the continuous model assumes that the attraction of criterion by memory trace, and memory trace by criterion occur with the same time constant. Are there theoretical and/or empirical considerations that would support this choice, or is it possible that the mutual attraction occurs at different time scales (e.g. memory trace drifts faster towards the criterion than vice versa)?

Furthermore, do the discretized and continuous models adequately capture the exponential decay of the history bias? One could use the models to simulate the rat's choices given their particular stimulus sequences and analyze that simulated data similar to what is shown in Figure 3e or f.

3. The author's findings add to an already substantial body of literature on adaptation and history biases. In the introduction, the authors state that "The magnitude of this repulsive perceptual effect in relation to the elapsed time between $n-1$ and n has not yet been mapped out [...]". I would disagree. Here are a few references which have investigated the strength of repulsive adaptation as a function of inter-stimulus interval:

- Measuring the strength of the TAE as a function elapsed time between adaptor ($n-1$) and test stimulus (n):

Magnussen, S., & Johnsen, T. (1986). Temporal aspects of spatial adaptation. A study of the tilt aftereffect. *Vision Research*, 26(4), 661–672. [https://doi.org/10.1016/0042-6989\(86\)90014-3](https://doi.org/10.1016/0042-6989(86)90014-3)

- Recovery time course from brief (<1s) adaptations to spatial contrast: Pavan, A., Marotti, R. B., & Campana, G. (2012). The temporal course of recovery from brief (sub-second) adaptations to spatial contrast. *Vision Research*, 62, 116–124. <https://doi.org/10.1016/j.visres.2012.04.001>

- Recovery time course from longer (>1s) adaptations to spatial contrast: Greenlee, M. W., Georgeson, M. A., Magnussen, S., & Harris, J. P. (1991). The time course of adaptation to spatial contrast. *Vision Research*, 31(2), 223–236. [https://doi.org/10.1016/0042-6989\(91\)90113-J](https://doi.org/10.1016/0042-6989(91)90113-J)

The influence of inter-stimulus interval has also been investigated for neural adaptation:

- In anaesthetized monkeys:

Patterson, C. A., Wissig, S. C., & Kohn, A. (2013). Distinct Effects of Brief and Prolonged Adaptation on Orientation Tuning in Primary Visual Cortex. *Journal of Neuroscience*, 33(2), 532–543. <https://doi.org/10.1523/JNEUROSCI.3345-12.2013> see their Figure 8.

- And in awake mice:

Jin, M., Beck, J. M., & Glickfeld, L. L. (2019). Neuronal Adaptation Reveals a Suboptimal Decoding of Orientation Tuned Populations in the Mouse Visual Cortex. *The Journal of Neuroscience*, 39(20), 3867–3881. <https://doi.org/10.1523/JNEUROSCI.3172-18.2019> see their Figure 1H.

All of these studies suggest an exponential decay of behavioral and neural adaptation across time. While this makes the current finding of the opposite effect perhaps even more intriguing, I think it's likely that the repulsion effect that the authors find in their task might be different in nature to the more classical stimulus adaptation investigated in the studies above. I would appreciate if the authors would take this literature into account, and discuss potential differences to these previous findings.

The authors further appear to suggest that nothing is known about the temporal dynamics of repulsion across longer sequences of trials (“[...] has not yet been mapped out [...] nor has the stimulus-driven repulsion over longer sequences of trials.”). Again, I think this is not entirely correct, and here are some references that could be discussed.

- Chopin, A., & Mamassian, P. (2012). Predictive Properties of Visual Adaptation. *Current Biology*, 22(7), 622–626. <https://doi.org/10.1016/j.cub.2012.02.021>

- Gekas, N., McDermott, K. C., & Mamassian, P. (2019). Disambiguating serial effects of multiple timescales. *Journal of Vision*, 19(6), 24. <https://doi.org/10.1167/19.6.24>

- Fritsche, M., Spaak, E., & de Lange, F. P. (2020). A Bayesian and efficient observer model explains concurrent attractive and repulsive history biases in visual perception. *ELife*, 9, e55389. <https://doi.org/10.7554/eLife.55389>

Finally, quite recently a few papers have been published on choice repetition effects in rats and humans (e.g. Lak et al., 2020; Mendonça et al., 2020). In the context of these papers, I find it somewhat curious that the authors do not find any effects of previous choices. I do not expect the authors to offer a comprehensive explanation of these apparent differences, but I do think that these studies should be mentioned, to contextualize the current findings.

Lak, A., Hueske, E., Hirokawa, J., Masset, P., Ott, T., Urai, A. E., Donner, T. H., Carandini, M., Tonegawa, S., Uchida, N., & Kepecs, A. (2020). Reinforcement biases subsequent perceptual decisions when confidence is low, a widespread behavioral phenomenon. *ELife*, 9, e49834. <https://doi.org/10.7554/eLife.49834>

Mendonça, A. G., Drugowitsch, J., Vicente, M. I., DeWitt, E. E. J., Pouget, A., & Mainen, Z. F. (2020). The impact of learning on perceptual decisions and its implication for speed-accuracy tradeoffs. *Nature Communications*, 11(1), 2757. <https://doi.org/10.1038/s41467-020-16196-7>

Minor comments:

4. Figure 2c: The only way in which the rat can figure out the stimulus range of the current session with certainty is by encountering a 28 mm/s or 163 mm/s stimulus, which are exclusive to the respective stimulus ranges. Have the authors compared sessions in which neither of these two stimuli were presented during the first 3 trials? If the curves would still be separated in this comparison, this may suggest that rats did not learn the stimulus range based on the current session, but rather inferred it from the previous session's range and the alternation scheme.

5. Figure 2b and c: Is there a specific reason to show p (“strong”) in the scatterplots? This metric is strongly correlated with the PSE. I would rather appreciate non-overlapping representations of the

bootstrapped PSEs (inset in panel c). The distributions appear to overlap quite a bit, but this is difficult to see when the datapoints are plotted on top of each other.

6. Figure 2e: While the curves are indeed separated, the rats show only weak adjustments to the two different stimulus distributions. When comparing Figure 2 d and e, it seems that it is only the PSE in the dark condition that shifts rightwards, whereas the PSE in the light condition remains put. Could this be evidence that rats do not use the full probabilistic representation of the stimulus distribution to adjust their criterion?

7. Perhaps it's just me, but what the authors call "dark green" in Figure 2 looks very blue to me. Perhaps adjust the labels to avoid confusion.

8. Supplementary Figure S1a: Could the authors show a similar plot for previous incorrect choices? Relatedly, if the bias would be driven by the previous choice rather than the previous stimulus, the slopes of the green and purple curves in Figure 3 should have opposite signs, right? Perhaps, the authors could point this out to further substantiate the role of the previous stimulus rather than choice.

9. Figure 3c: I am not entirely convinced that it is justified to state that the bias is mostly independent of the outcome of trial n-1. Visually, the slopes and/or asymptotes appear to differ between previous correct and incorrect trials. In order to shed more light on this, the authors could run a model-based analysis. Specifically, the authors could fit the data with logistic functions (including "lapse" rates), quantify the difference in slopes, and compare this empirical difference with a permutation distribution of slope differences (permute the labels of previous correct/incorrect trials, re-fit logistic models, record slope difference, repeat 1000 times). Is the slope significantly steeper after previous correct trials? Are the asymptotes significantly different? This analysis would provide a more detailed view on the role of the previous outcome/reward.

10. Supplementary Figure S2b: "Regression slopes for $p(\text{judged "strong"}) \times \Delta \text{speed}$ in trial n-1, for each trial n stimulus value." I don't understand. What is " $p(\text{judged "strong"}) \times \Delta \text{speed}$ "? To look at the effect of the previous stimulus, I would expect to look at the effect of the $\Delta \text{speed}_{n-1}$ regressor.

Supplementary Figure S2c: I am not able to follow the analysis here based on the description in the legend. Is the plot in panel c a depiction of current PSE as a function of current mean of the Gaussian cdf, color coded by previous speed on correct trials? I would appreciate if the authors would elaborate on the rationale of this analysis.

11. Figure 4d: It took me a while to understand that the two ROC curves are plotted on top of each other, since the blue area looks like a border of the grey area. It does make sense now, but I am wondering whether there would be a clearer way of showing that these are two overlapping areas and not just one grey area with a blue border.

12. Figure 5c and d: From the figure caption, I take that the human data was pooled and then split according to grand median ITI. This raises the possibility that differences in adaptation strength may be driven by between-subject variability in response times rather than between-trial variability. Does the regression pattern look similar when splitting ITIs per participant and then analyze the history effect based on these splits? Similarly, were the ITI quartiles for the rat data computed per rat or for the pooled data?

Figure 5e: It would be more informative if the datapoints would be color coded for rat identity, rather than redundantly coding the ITI. This would allow to see whether there might be systematic differences between rats (e.g. Rat 12 had quite fast ITIs overall, Rat 9 was quite slow – is the trend of the ITI repeated within each rat, or mainly driven by between-rat differences)?

13. Do the human participants show effects of their previous choice or feedback? How does the repulsion bias decay over trials. Currently, there is only one figure panel showing the human data. While panel shows the replication of the most important finding, I think it would be nice to show that the human biases are comparable to the rat biases in other ways too (i.e. only effects of previous stimulus, not choice or reward; exponential decay across trials). This could be a supplementary figure.

14. P. 19, line 521: "The AUC of the ROC for the continuous, time-dependent model was significantly greater than that of the discretized model (Figure 6c), demonstrating the continuous model's more accurate predictions."

I would not use the term "significantly" to avoid confusion, as no inferential static is provided.

15. P.26, line 712. "Contraction bias might be explained if the interaction between the short term buffer and μ is symmetric (reciprocating arrows in Figure 7b)."

It is not clear to me why the interaction would have to be symmetric. It would be helpful if the authors could elaborate on this.

Reviewer #3:

Remarks to the Author:

This manuscript by Hachen et al. aimed to dissect how different aspects of trial history, such as stimulus, choice, and reward, may affect current trial decision, by training rats and humans to perform a tactile categorization task. Based on the results, the authors concluded that previous stimulus, but not choice, influences current decision, and this effect tends to intensify as inter-trial-interval prolongs. A simple model is able to capture this history-dependent behavioral effect.

This is an intriguing study. However, I have the following comments for the authors to consider and to make certain that the claims are valid conclusions of the evidence.

Major comments:

In Fig. 2a, to study how rats' performance evolves in a single session, the authors constructed psychometric curves from the first 5 trials in each session and compared it to a shuffled control. What is the rationale of choosing the first 5 trials, given that each session consists of several hundred trials and 9 different stimulus levels? It appears to me that a more direct approach would be to systematically examine how psychometric curves evolve over time. A comment of the similar nature applies to Fig. 2b and c. The observation that the relative positions of the two psychometric curves don't change when all trials are included or only the first 3 trials are included does not warrant the conclusion that the previous session exerts no effect on the performance of the current session. Additional analyses are needed to support this claim. It also seems arbitrary to use the first 5 trials for one analysis, and the first 3 trials in another. In addition, the results in Fig. 2c seem to have large variability. To what extent are the separations of the two curves significant? Is this separation consistent across individual rats?

Fig. 2d and e tried to understand how rats solve this task, setting an internal threshold or detecting stimulus distribution. The results are intriguing and confounding at the same time. The authors showed that 1. Psychometric curves don't change when shifting the stimulus boundary, and 2. Psychometric curves do change when shifting both the boundary AND the distribution. I don't think these findings warrant the conclusion that rats detect stimulus distribution but not threshold/boundary. In addition, I find it difficult to comprehend the statement that 'rats detect stimulus distribution'. What does it mean? Why does their PSE follow the mean of the distribution?

Fig. 3 attempted to dissect the effects of previous stimulus amplitude and choice on current decision. The authors concluded from 3b that previous choice does not affect current decision. However, 3b only considered rewarded choices. It is not clear to me why unrewarded choices were excluded from this analysis. It is also unclear to me why 3b only used 0 previous speed and 3c used all previous speeds. If choice/reward history does not affect current decision, how did these rats learn to perform the task?

The authors showed that 1. stimulus distribution determines rat choices, 2. rat behavior is different in the first 3 trials using high and low ranges, and 3. the optimal trial history is 1-3 trials for the model. So, how are the first 1-3 trials enough for the rats to establish their internal reference and correctly perform the task?

Fig. 5 shows that longer ITI exerts stronger stimulus effects on decision. This is a bit surprising as we'd normally think such effects decay over time. I wonder whether there are other factors that may underlie this effect. For example, are ITIs correlated with task difficulty? Since trials are self-paced, are ITIs associated with the motivational states of the rats? The authors mentioned that they excluded the longest 20% of ITIs as they were associated with task unrelated behaviors. Are the remaining 80% completely free of such behavior?

Finally, a major missing piece in this work is an effort to reveal the neural mechanisms underlying the described behavioral effects, such as stimulus dependency and ITI dependency. Uncovering some aspects of the neuronal correlates would significantly enhance the readers' appreciation of the behavioral results and contribute to our understanding of history dependent perceptual judgement.

Minor comments:

Fig. 1b is confusing. The slope at inflection point is plotted in gray, and the Gaussian is also gray. Gray Y-axis is not labeled and it's unclear whether it is designating y-values for the Gaussian or the inflection point or both.

In a number of figure panels, axis values are not aligned to tick marks, reducing readability.

In Fig. 2e, why PSE = 118 corresponds to a 40% performance, if the definition of PSE is 50% performance?

Evidence detailed in Fig. 3 e and f suggests trial history is strongest for n-1, but the discretized model used $\tau = 3.3$ trials for the trial history. Some discussion regarding this difference is needed.

Line 627-628: It is suggested that no terms are required for more distant trials other than n-1. What is the justification? No evidence was shown for the updated model that suggests trial n-1 is the optimal number of trial history. The discretized model used $\tau = 3.3$ trial, for example.

The models discussed in this paper did not include noise as a factor. Biological systems are subject to both intrinsic and extrinsic noise. Although not necessarily detrimental to the study, including noise (either for the sensory memory trace or the decision criterion) may improve model performance.

Fig. 6c: Is the Y axis miss labelled? Otherwise, is it meaningful to compare 2 measures when their AUC difference is on the order of e^{-15} ? In addition, are the predictions in 6c an average of all speeds? It is important to compare model predictions in the 0 speed condition.

Major stylistic revisions are needed to enhance readability.

REVIEWER COMMENTS

Reviewer #1

Point #1

Reinartz et al consider the fundamental question of how perceptual decision making depends on the prior history of stimuli, rewards and choices. This is a topic of considerable current interest and this study advances our understanding by applying quantitative analysis to data from rats and humans trained to discriminate the speed of a tactile vibration. The main claims of the study are that the primary history effect is a stimulus repulsion effect from the previous trial, that both humans and rats show the repulsion and that the data can be accounted for by a remarkably simple model.

Overall, I find the technical quality of the work to be high and the claims to be convincing. An important strength is that the authors have designed a task that delivers 1000s of trials; fundamentally it is this large data volume that permits them to tease apart effects of variable type, time course etc that previous studies knew to be potentially important but lacked the statistical power to resolve. The field needs these quantitative insights. Another strength is that the models used are simple, with very few parameters. This is crucial since it means that the models do not just replicate the data but add insight into what the key parameters that drive performance. A final strength – which separates this study from most others - is the evidence presented (from the human work) that the basic finding here is not merely a quirk of the rat (or indeed of the human) but a general principle of perception. Overall, the paper is likely to have significant influence on the broad community of researchers interested in perceptual decision making and its neural basis.

Authors' reply

Thank you for your appreciation of the work.

Reviewer #1

Point #2

Medium:

1. There is much current interest in the idea that normative (especially Bayesian) theories are useful for understanding perception. On the face of it, perceptual repulsion (and the models proposed here, eg equation 3) are suboptimal given the task, and therefore present a challenge to normative theories. It would be useful to discuss this: do the authors think that Bayesian theory is challenged by their findings?

Authors' reply

Our experiment represents a special case of categorization: a fixed range and a fixed boundary (within session, at least). In this special case, reward boundary is equivalent to the mean of all past stimuli. According to a strict normative approach (e.g. Behrens et al., 2007; Gallistel et al., 2001), any repulsive or attractive history effect in a task with a fixed boundary would be suboptimal. To solve the task, the perceptual/decision making system may pose the problem: Is the current stimulus stronger or weaker than the average of all past stimuli? Memory limits do not permit a stored representation of all past stimuli, but only of some limited sequence of past stimuli, as dictated by τ .

But an alternative perspective might focus on the very fact that the fixed range and boundary is a special case. In natural ecological settings, boundaries are constantly changing, e.g. the call of a predator may be judged as “strong” against of a background of innocuous sounds, but that same call may be “weak” a moment later when compared to other calls from a larger ensemble of predators. A flexible mechanism to assess the sensory environment is advantageous. Thus, encountering the changeable world, the perceptual/decision making system’s task might be: Is the current stimulus stronger or weaker than *recent* stimuli? Seen in this way, the mechanism quantified by our repulsive model is in fact a Bayesian, normative framework for defining *the more immediate past*, by virtue of its exponentially boosted weighting of stimuli by temporal order.

Perceptual/decision making systems may have evolved built-in mechanisms for adjusting to changing stimulus statistics. This argument is compatible with Bayesian theories of perception that assume built-in priors (e.g. Mathys et al., 2014). In our data set, the individual differences in time

constant τ (Figure 6d) would thus be interpreted as individual differences in how volatile a world the brain is programmed for.

To summarize the considerations above, we have added the following Discussion (lines 706-713):

It must be kept in mind, however, that the long τ that is advantageous for the fixed boundary condition of the present task (approaching infinity in the ideal case) is not necessarily ideal for a volatile stimulus context, for example if the stimulus distribution and reward boundary were programmed to move every 10 trials. In the latter case, a decision boundary built on the history of only recent stimuli may be normative if one assumes that perceptual-decision making systems have selected for mechanisms that allow them to rapidly adjust to volatile environments, where stimulus range and categorical boundary may shift unexpectedly. Adaptation to this sort of environment could be implemented in the brain by hierarchical Bayesian mechanisms^{61,62}.

Behrens, T. E. J., Woolrich, M. W., Walton, M. E. & Rushworth, M. F. S. Learning the value of information in an uncertain world. *Nature Neuroscience* **10**, 1214–1221 (2007).

Gallistel, C. R., Mark, T. A., King, A. P. & Latham, P. E. The rat approximates an ideal detector of changes in rates of reward: implications for the law of effect. *J Exp Psychol Anim Behav Process* **27**, 354–372 (2001).

Mathys, C. D. *et al.* Uncertainty in perception and the Hierarchical Gaussian Filter. *Front. Hum. Neurosci.* **8**, (2014).

Reviewer #1

Point #3

2. I buy that, in their task, perceptual repulsion is the major history effect. However, in Discussion, the authors state the effect of choice and reward are “negligible”. I am not fully convinced by this. First, there is some systematic separation between the blue and red curves in Fig 3b, indicating some modulation by n-1 choice.

Authors' reply

The authors recognize that “negligible” is too absolute a term and we have replaced it with a more neutral diction (see excerpt below).

The red and blue curves of Figure 3b deviate slightly from complete overlap for trial n stimuli of lowest Δ speed, -2 to -4. After mining multiple statistical procedures (see below), we were unable to confidently rule out the null-hypothesis regarding their difference. In other words, we cannot be confident that the apparent differences did not occur by chance.

First, we performed a bootstrap test with the psychometric parameters of the two curves in Fig. 3b (see Figure S4a), which did not support the hypothesis of a difference (even at a relatively permissive confidence level of 90%). Such an analysis, where the two sets of data are separated conditional on the $n-1$ choice, is the most reliable test.

As a further attempt, we performed a logistic regression analysis including main effects and interactions of previous stimuli and previous correct choices (see Reply Letter Table 1 below). Although the analysis returned a statistically significant negative coefficient for the main effect of choice $n-1$ on choice n , the extreme collinearity between stimulus and choice predictors ($R=0.88$; variance inflation factor ~ 5) does not allow us to draw a conclusion about the previous choice's main effect. In short, when considering all possible stimulus values for $n-1$, any “secondary” effect of the previous choice would be highly confounded by the previous stimulus's overwhelming “primary” effect.

In summary, although speculation as to the cause of a possible effect of previous choice would be interesting, we believe that the evidence in Figure 3b for separation of trial n choices according to the trial $n-1$ choices is not well-enough grounded to support such speculation.

Nonetheless, we have modified the text to be more cautious both in Results (lines 236-241):

If trial n choice were related to the preceding action, the two curves would be significantly

separated. Though there may be a trend for trial $n-1$ choice to affect the subsequent choice when trial n stimuli were weakest (Δ speed -2 to -4), curve parameters were not statistically different (bootstrap test, 90% confidence level; see Figure S4a). Instead, the overlap in confidence intervals argues that the trial n choice was influenced only mildly, or not at all, by trial $n-1$ action or choice.

and in the Discussion (lines 627-629):

Trial-by-trial analysis revealed the effects of previous choice (Figure 3b) and reward (Figure 3c) to be minor, leaving the physical magnitude of the past stimuli as the major factor responsible for variability in perceptual judgments in the given task.

As to statistical measures, the logistic regression model with interactions between previous stimulus and previous choice effect are the following (in Wilkinson notation):

$$p(\text{judged } strong)_n \sim 1 + \text{Stimulus}_n + \text{Stimulus}_{n-1} * \text{Choice}_{n-1}$$

Reply Letter Table 1

	Estimate	SEM	p-value
Intercept	0.1191	0.0157	<0.001
Stimulus _n	0.3390	0.0023	<0.001
Stimulus _{n-1}	-0.0352	0.0056	<0.001
Choice _{n-1}	-0.1763	0.0225	<0.001
Stimulus _{n-1} :Choice _{n-1}	-0.0157	0.0081	0.051

Reviewer #1

Point #4

Second, there is also systematic separation between the green and purple curves in Fig 3c, indicating some modulation by reward. I do not dispute that these effects are most likely minor, but given that there does seem to be some effect, it would be interesting to take advantage of the authors' modelling framework to quantify them. In particular, the model of Equation (3) predicts rat choice on 78% of trials. It would seem that the model (and/or their GLM) could

straightforwardly be generalised to include choice and/or reward terms. This would allow numbers to be attached to the benefit (or not) of including choice/reward.

Authors' reply

Differently from the case of Figure 3b, in Figure 3c the difference between the two curves seems to be more systematic. We pursued further statistical tests and found a minor but significant effect. Specifically, the requested GLM:

$$p(\text{judged } strong)_n \sim 1 + \text{Stimulus}_n + \text{Stimulus}_{n-1} * \text{Reward}_{n-1}$$

reveals that previous reward interacted weakly with stimulus history. Specifically, the sign of the coefficients reflects a decreased influence of the previous stimulus following incorrect trials. That decreased influence explains the flatter shape of the purple curve with respect to the green curve (supplementary table T1, also copied in here as Reply Letter Table 2):

Reply Letter Table 2

	Estimate	SEM	p-value
Intercept	0.0299	0.0093	0.001
Stimulus_n	0.3273	0.0020	<0.001
Stimulus_n-1	-0.0245	0.0046	<0.001
Reward_n-1	-0.0189	0.0106	0.075
Stimulus_n-1:Reward_n-1	-0.0461	0.0049	<0.001

Additionally, we performed a bootstrap test with the psychometric parameters of the two curves in Figure 3c following correct and incorrect trials (a procedure similar to the investigation of the effect of previous choice under point #3; also see Figure S4b). The test did not exclude a null difference in midpoint and slope (confidence level < 90%), however it evidenced a difference in the asymptote parameters. Both lapse parameters, γ and λ , seemed to decrease after an incorrect trial (confidence = 92%, 97%, respectively). Therefore, the interaction between previous stimulus and previous reward, estimated by the GLM, could be explained at a finer level as an increase in

lapse probability following incorrect trials.

The interpretation is complex. The apparent attractive effect could be caused by an increased probability to lapse or to switch side following incorrect trials, as suggested by the γ and λ parameters. However, an additional factor might be at work. The sensory-perceptual process on trial n might be conditioned by a non-veridical sensory representation after an incorrect (unrewarded) trial $n-1$. If stimulus $n-1$ was weak but incorrectly judged as strong, why did the error occur? It is possible that $n-1$ was erroneously encoded by the sensory system as stronger than its mean expected value (due to noise or fluctuations); the trial n representation would then be *less repulsed from weak* as a consequence of $n-1$ being erroneously encoded as strong.

Above, we depict just two possible explanations of the small separation between curves of Figure 3c – a higher-order effect linked to lapsing or switching and a lower-order effect linked to sensory coding dynamics. Until one knows all possible causes of error, and which cause was at play on individual trials, it is difficult to hazard an interpretation of the separation between the two curves.

The modified text is the following (lines 265-272):

A logistic regression model including an interaction term between previous stimulus and previous reward was fitted to the data. The interaction weight was statistically significant ($p < 0.001$; see Supplementary Table T1), indicating that the repulsive effect of stimulus $n-1$ was slightly reduced if that trial's choice was not rewarded. Bootstrapping the parameters of the psychometric curves from Fig. 3c (see Figure S4a) revealed this phenomenon to be mainly driven by an increase in lapse probability following an incorrect trial, suggesting a slight lose-switch tendency which might partly counterbalance the effect of previous stimulus after incorrect trials.

Reviewer #1

Point #5

Minor:

In a few places, the reasoning is hard to follow. Please clarify lines 258-9 and, particularly, 271-8.

In line 203-4, it is not clear why the result of Fig 2e cannot be due to a change in rule boundary.
Line 131 – what is the basis for this assertion?

Authors' reply

Regarding lines 258-9, the wording "...we could not reject the null hypothesis as regards the effect of choice (Figure 3b) or reward (Figure 3c) associated with that trial" was an indirect way of saying that the apparent differences between the curves did not reach statistical significance. In light of Reviewer #1's points 3-4, we have changed the analyses, figures, and descriptive text.

Regarding lines 271-8, first, we have highlighted that the black dashed line in Figure 3a gives the trial n PSE with all trials merged, unconditional on $n-1$ Δ speed, a point that was missing in the original submission (lines 218-224, new text underlined):

Figure 3a shows the psychometric curves for trial n , for all rats merged, sorted according to Δ speed of trial $n-1$. This sorting resulted in a spread across the curves – when trial $n-1$ presented a high Δ speed stimulus, the probability of trial n being categorized "strong" decreased; when trial $n-1$ presented a low Δ speed stimulus, the probability of trial n being categorized "strong" increased. The effect was graded for intermediate values of trial $n-1$ Δ speed. The PSE computed with all trials merged, unconditional on $n-1$ Δ speed, is given by the black dashed vertical line.

Next, we have rewritten the related section as follows (now lines 290-296):

Given the mainly horizontal shift induced by stimulus $n-1$, the PSE constitutes a robust measure of the history-dependent bias. On the data set averaged across rats, 9 values of the PSE associated with 9 values of $n-1$ Δ speed are shown as the blue squares in Figure 3a. These can be considered graded quantities of history-dependent curve shift, or bias. When the same procedure is carried out for 6 individual rats, 6 data points are associated with each value of Δ speed on trial $n-1$ (Figure 3d, left panel). The size of the trial n bias was linearly correlated with trial $n-1$ Δ speed (Pearson's $R = 0.869$; $p < 0.001$). This linear correlation quantifies the $n-1$ -dependent curve shifts seen in Figure 3a.

Regarding lines 203-4, the reviewer states that it is not clear why the result of Figure 2e cannot be due to a change in rule boundary. If the rats' choices were sensitive to the boundary *per se*, the curves of Figure 2d would have been separated as a consequence of boundary change. The difference between the experiments illustrated in Figure 2d and 2e is related to neither the overall stimulus range nor to the category boundaries (indeed, these are matched in 2d and 2e). The only difference is the extraction probability of each stimulus speed, which is uniform in the experiment of Figure 2d but balanced by category for the experiment of Figure 2e.

Next, the Reviewers asks what is the basis for line 131, which reads: "Since rats appear to establish the relationship between *sp* and category at the onset of every session," and continues as "we asked how they would adjust their choices in response to changes in range and category boundary." We agree that Figure 2a does not provide strong enough proof of the rat establishing the relationship between *sp* and category at the onset of every session, and have deleted that part of the sentence. It now reads "We asked how rats would adjust their choices in response to changes in range and category boundary." (line 130)

Reviewer #1

Point #6

Line 170. The data points of Fig 2c are quite noisy (necessarily, since based on few trials) – can the authors exclude the null hypothesis that the separation shown could arise from chance?

Authors' reply

The reviewer is correct; the bootstrap test between the two conditions does not exclude a null difference either for PSE or for p ('strong'), even at the 90% confidence level.

As an additional analysis, we tested the evolution within the session of the separation between psychometric curves by progressively adding trials, starting from the onset of the session. For each set of trials, 1 to n , we compared the individual-session PSEs across the two conditions. A significant separation ($p < 0.05$) in PSE and p ('strong') arises about 30 trials into the session. While the curves that begin to form and separate in Figure 2c are consistent with what happens later in the session, it is not quantitatively justified to make a claim. We are grateful for this clarification.

We have reinterpreted the findings now to emphasize a different, but related point: that the lack of significant separation means that the curve separation from the preceding session was not carried over. The relevant text, completely rewritten, is now (lines 165-176):

To evaluate the time course of this adjustment, Figure 2c shows the average psychometric curves of rats across the first 3 trials of every session, yielding two main findings. First, at the session onset the rats showed low performance (shallow logistic functions and high lapse rates), consistent with the results of Figure 2a. Second, the absence of significant separation between the two curves suggests that the psychometric function of the preceding session was not carried over. If it were carried over, the blue curve would have been displaced to the left of the light green curve, inasmuch as the current low-range session would initiate with a choice function from the preceding high-range session, and vice versa. Further analyses showed that the separation between the curves became statistically significant after about 30 trials (rank-sum test between PSEs, $p < 0.05$; $p < 0.001$ after about 50 trials). In conclusion, rats initiated the session with poor performance and without any observable residual influence of the previous session's range and/or boundary; their behavior adapted over time to the current session.

Reviewer #1

Point #7

Line 41. Other work, not cited, is relevant (Campagner et al, 2019; J Neurosci) since concerns rodent tactile decision-making, finding choice repetition effect.

Authors' reply

Thank you for pointing out this citation, which is related to our study; we have inserted it into the Introduction (lines 39-41):

When compared directly within the same experiment, the most recent choice (on trial $n-1$) is found to exert an *attractive* effect on the current choice (trial n); that is, choices tend to be repeated^{8,16,18,19}.

Reviewer #2

Point #1

The study investigates the temporal dynamics of history biases in perceptual decision-making in rats and humans. The authors find that the recent stimulus history exerts a repulsive bias on current perceptual decisions, akin to well-known repulsive adaptation biases. Surprisingly, the authors find that the influence of the previous stimulus increases as the time between previous and current stimulus increases. Intuitively, I would have predicted to see the opposite pattern, and to my knowledge such an increase in history biases with increasing inter-trial intervals (ITIs) has not been reported before. The authors propose a new model in which the decision criterion is gradually updated by each new stimulus. The manuscript is well written and the experiments appear thorough. In my opinion, the study presents an important addition to a rapidly increasing body of literature on history biases in perceptual decision-making. While I don't see any grave issues with the study, I have a few comments and suggestions that would need to be addressed before recommending the study for publication.

Authors' reply

Thank you for your appreciation of the work. We agree that the increase in history bias with increasing ITI is surprising and novel and is thus one of the findings that will be of interest to readers.

Reviewer #2

Point #2

1. The authors show an intriguing pattern of increasing repulsion from the 1-back stimulus as the ITI between current and previous stimulus lengthens. The authors also find that 2- to 5-back stimuli still exert repulsive biases. I was wondering whether the authors have investigated

whether those biases were also modulated by the current ITI. If I am not mistaken, their continuous model would only predict (/capture) modulations of the 1-back repulsion. I am asking this, because it is not entirely clear to me whether the variability in the ITI would only affect the integration of 1-back stimulus information with the decision boundary, or may also affect how the current stimulus is processed (e.g. rats/humans might be less alert following a long ITI; or ITIs might be longer in periods of inattention; or ITIs might increase over the course of the session and thereby correlate with attention) thereby potentially boosting the influence of multiple non-stimulus factors. I would suggest that the authors could run a GLM analysis similar to the one shown in Figure 3f, separately for current short and long ITIs, or add the current ITI as an interaction with previous stimulus evidence.

Authors' reply

"If I am not mistaken, their continuous model would only predict (/capture) modulations of the 1-back repulsion."

No, this is not the case, because earlier stimuli (prior to $n-1$) determine the decision criterion's starting value at the moment of presentation of $n-1$. The decision criterion essentially holds a memory trace of stimulus history in its current value.

The continuous model captures the empirical finding that a long ITI following stimulus $n-1$ increases the $n-1$ repulsive effect and thereby partially overwrites the biases accumulated from stimuli $n-2$, $n-3$, and so on. To make this clearer, we have carried out a new analysis and created a new figure as described in lines 550-557:

To investigate how the trials prior to $n-1$ would be expected to influence trial n choices according to the two models, we allowed both the discrete and continuous models, with τ already fitted, to predict rats' choices for the entire set of trials in their real temporal order. From the resulting series of choices, we used the GLM first employed for Figure 3f to solve for the weighting of the β coefficients (Eq. 2). In Figure 6c, it can be seen that for both models the coefficients decay with an exponential trend. Because the analysis pools trials with different preceding ITIs, the continuous and discrete models converge. For comparison,

the GLM coefficient weights fitted on the observed rat data are shown in gray, carried over from Figure 3f.

The reviewer is correct in predicting that the bias of the 2- to 5- back stimuli is modulated by the current ITI. We performed a logistic regression analysis with interaction terms between the effect of stimulus $n-2$ and two adjacent ITIs (supplementary table T2, also copied in here as Reply Letter Table 3; ITI _{$n-1$} is the ITI bin between trial $n-1$ and trial n , and ITI _{$n-2$} is the ITI bin between trial $n-2$ and trial $n-1$; we considered only previous correct trials). As shown in previous analyses, stimulus $n-2$ exerted a repulsive effect on choice n (although weaker than the effect of stimulus $n-1$); this effect was enhanced with increasing ITI from stimulus $n-2$ to $n-1$, but reduced with increasing ITI from stimulus $n-1$ to n (interaction terms' p-values ≤ 0.01).

Reply Letter Table 3

	Estimate	SEM	p-value
Intercept	0.0201	0.0089	0.0239
Stimulus _{n}	0.3590	0.0039	<0.001
Stimulus _{$n-1$}	-0.0713	0.0033	<0.001
Stimulus _{$n-2$}	-0.0312	0.0122	0.0108
Stimulus _{$n-2$} :ITI _{$n-1$}	0.0169	0.0069	0.0136
Stimulus _{$n-2$} :ITI _{$n-2$}	-0.0185	0.0069	0.0072

We have modified the text as follows (lines 451-455):

Finally, a GLM analysis (Supplementary Table T2) uncovered a significant repulsive effect of stimulus $n-2$, which was reduced with increasing ITI from stimulus $n-1$ to n . This is explained by the memory of $n-2$ being “written over” as $n-1$ exerts a progressively stronger effect. Accordingly, with reduced ITI from stimulus $n-1$ to n the effect of $n-2$ was larger.

The reviewer raises the concern that ITIs might be longer in periods of inattention, or ITIs might increase over the course of the session and thereby correlate with attention. Unfortunately, there is no evidence in our study or in any study known to us as to what causes variability in ITI; it may reflect attention, but we have no means of directly monitoring attention. However, we have now

checked whether ITIs shift systematically and found that they are quite stable. Furthermore, the effect of ITI on trial n choice for the full session is also quite stable. The new analyses are consistent with the history effect on decision criterion not depending on the behavioral state, as outlined in lines 444-451:

We performed control analyses excluding potential confounds (see Supplementary Figure S6). First, we replicated Figures 5b and 5c after excluding previous incorrect trials (Figure S6). Furthermore, using only the first 150 trials of each session (when motivation or urgency might be stronger), the ITI had the same effect on trial n choice as for the full session. There was a difference of just 50 ms in median ITI for the first 150 trials of each session compared to the whole session (7.57 s versus 7.62 s), suggesting that the pace of the task was stable within sessions. The difficulty of trial $n-1$ (i.e. the closeness of $\Delta speed$ to the category boundary) did not affect the ITI from $n-1$ to n (Figure S6c).

Reviewer #2

Point #3

It would be also informative to test whether lapse rates or slope of the psychometric functions correlate with ITI, to rule out that effects of ITI reflect changes in current stimulus processing, rather than changes in the integration of decision criterion and 1-back stimulus information.

Authors' reply

In order to investigate whether lapse rates of the trial n psychometric functions correlate with $n-1$ to n ITI, we performed a more sensitive analysis, rather than relying on the fitted values of the parameters (since lapses and midpoint tend to be highly correlated). In new Supplementary Figure S5, we have considered two models of the effect of ITI: one in which ITI affects psychometric curve lapse rate (Supplementary Figure S5a, left plot), and one in which ITI affects psychometric curve horizontal position but *not* lapse rate (Supplementary Figure S5b, left plot). The absolute difference between long-ITI psychometric curves and between short-ITI psychometric curves for the two models are given in Supplementary Figure S5a-b (right plots). If ITI affects lapse rates, then the difference between long-ITI and short-ITI psychometric curves will be similar across all values of $n \Delta speed$ (Supplementary Figure S5a, right plot); if ITI shifts curves horizontally rather

than through lapse rates, then the difference between long-ITI and short-ITI psychometric curves will be expanded near Δspeed of 0 and compressed near Δspeed of -4 and 4 (Supplementary Figure S5b, right plot). The observed data are shown in Supplementary Figure S5c, and can be seen to correspond to the hypothetical case of midpoint (but not lapses) being modulated by ITI.

The relevant text (lines 429-433) is:

Next, we assessed how stimulus $n-1$ modulated the trial n psychometric function parameters in relation to the ITI from $n-1$ to n . The slope of the psychometric function was not significantly modulated by ITI (interaction test between $n-1$ Δspeed and ITI, $p = 0.47$; ITI's main effect $p = 0.18$). Further analyses (Figure S5) indicated a significant influence of ITI on psychometric function midpoint but not on lapse rates.

Within the same point, the Reviewer is suggesting that our model is invested in demonstrating that $n-1$, over the course of the ITI, affects the decision criterion as opposed to current stimulus processing. In fact, our interpretation is open to both possibilities. In Figure 7a of the original and revised submission, both possibilities are presented, along with the text, excerpted below (lines 667-678):

Two ways in which the brain may mediate the stimulus-to-decision mapping⁵⁶, are illustrated in Figure 7a... In the left panel... the criterion itself is history-dependent, moving downwards after a weak $n-1$ and upwards after a strong $n-1$...

An alternative form of stimulus-to-decision mapping is illustrated in the right panel of Figure 7a. Here, stimulus n evokes a sensory response that varies according to stimulus $n-1$.

Reviewer #2

Point #4

2. The authors use cross-validation to show that their continuous model outperforms the discretized and “no history” models. However, I think that the authors could do a better job in showing that their model actually provides an adequate fit to the empirical data. For instance, the continuous model predicts that as the ITI goes to infinity, the decision criterion should converge to

a value halfway between the initial criterion and the previous speed. In Figure 5e, it looks like the effect of the previous stimulus plateaus rather quickly after 7 or 8 seconds. Does this reflect the asymptote which is halfway between initial criterion and the previous speed?

Authors' reply

We appreciate this observation, and we concluded that Figure 5e is not the optimal representation of the data due to the distortions caused by normalizing the diverse τ across different subjects to a single time scale. We have removed the panel because it misleadingly suggested reaching an asymptote after 7-8 seconds (as the Reviewer reasonably interpreted it) and this is not the actual outcome of the model. In fact, for a τ of 40 s (expressed by some rats in Figure 6d), the decision criterion would reach 90% of the distance to the asymptote at 46 s, which is far beyond any ITI in the data set. The model's performance is further addressed under point #7. To further highlight the model's characteristic noted by the reviewer, we have added the following underlined sentence (lines 496-499):

Reworking these two equations, or using symmetry considerations, one can see that when t goes to infinity, $\mu(t) = (\mu(t_0) + \Delta speed(t_0))/2$; that is, the decision criterion would asymptotically reach a value midway between its previous value, $\mu(t_0)$, and the $n-1$ stimulus.

Reviewer #2

Point #5

In Figure 6b, which shows the model's prediction together with the data, the x-axis is cut off at 7.5 s or so (i.e. just around the beginning of the plateau). How does the fit look when extending the x-axis to 10 seconds (like in Figure 5e)?

Authors' reply

We are grateful for the Reviewer pointing to an error in the plot. It was mislabeled as ITI while the actual abscissa is the time from the $n-1$ go cue to trial n nose poke, not the entire ITI, which for the

last bin would be 9.2s (median). For better clarity, we changed the corresponding axis to provide information about the entire ITI.

Reviewer #2

Point #6

Relatedly, the continuous model assumes that the attraction of criterion by memory trace, and memory trace by criterion occur with the same time constant. Are there theoretical and/or empirical considerations that would support this choice, or is it possible that the mutual attraction occurs at different time scales (e.g. memory trace drifts faster towards the criterion than vice versa)?

Authors' reply

We fully subscribe to the Reviewer's focus on whether mutual attraction is symmetric or not. Indeed, figuring this out is one of the laboratory's major ongoing efforts. Let us refer to the time constants for convergence of the long-term buffer (i.e. decision criterion) towards the short-term buffer (i.e. memory trace) and short-term towards the long-term buffer as τ_{long} and τ_{short} , respectively. The manuscript's model describes the case where short-term and long-term buffers converge symmetrically. That is, $\tau_{long} = \tau_{short}$, and this single value of τ is a fixed property of a given rat's brain.

Two other possibilities exist:

(1) $\tau_{long} \neq \tau_{short}$ but the two, asymmetrical values of τ are fixed across different tasks and stimulus statistics,

(2) τ_{long} and τ_{short} vary in a task-dependent and stimulus statistics-dependent manner.

For instance, with regard to the 2nd possibility, in natural environments, the statistics of the sensory world are not always stable. "Volatile" environments comprise frequent changes in statistics. In reference memory, a large τ_{long} is ideal if the stimulus statistics and the rule boundary are fixed. A small τ_{long} is adaptive in a volatile environment, since the long-term buffer (the decision criterion) needs to shift rapidly.

In the Discussion, we note that the model's short term buffer might correspond to the working memory in a delayed comparison task. Correspondingly, the convergence of the short term buffer and long term buffer mimics a well described phenomenon in working memory research, contraction bias (Raviv et al., 2014) Also in working memory, there are advantages in having a flexible τ_{short} . If, within a session, the base stimulus varies widely from trial to trial, it is advantageous for τ_{short} to be very large, since this would make the short-term buffer more stable (less strongly attracted to the long-term buffer, minimal contraction bias) and boost the retention of the base stimulus. If, in a different session (or half of the same session), the base stimulus is fixed on every trial and only the comparison stimulus varies, it is advantageous for τ_{short} to be very small. This is because the base stimulus will overlie the long-term buffer, and a small τ_{short} would make the short-term buffer adhere more closely to the long-term buffer, boosting the stability of the base stimulus memory trace.

Thus, we are testing reference memory rats with varying degrees of stimulus statistical volatility. We are also training single rats to do two tasks (working versus reference memory) to investigate whether (and if so, by what rules) τ_{long} and τ_{short} vary according to the task and the ongoing stimulus statistics.

We limited the model at present because (i) with even a single value of τ it has good predictive power; the model would have a better fit but would be less powerful if allowed to fit two different values of τ . (ii) we do not yet have the basis for an independent test of the model's two different values of τ . What would be required is to tap into the two buffers and measure their time constants, which is the object of current research.

While the reviewer has clearly hit upon a theme that is central to our own thinking, the discussion of τ symmetry and τ flexibility would extend the paper and might become too speculative with the given data set. We have added a sentence in the Discussion (lines 754-758):

The current form of the model considers the strength of attraction between the decision criterion, $\mu(t)$, and the stimulus $n-1$ memory trace to be equal, causing the short-term and long-term buffers to converge symmetrically. Future studies will explore the possibility that

the attraction is asymmetric; indeed, the interaction between $\mu(t)$, and the stimulus $n-1$ memory trace might even be task-dependent.

Raviv, O., Lieder, I., Loewenstein, Y. & Ahissar, M. Contradictory Behavioral Biases Result from the Influence of Past Stimuli on Perception. *PLOS Comput. Biol.* **10**, e1003948 (2014).

Reviewer #2

Point #7

Furthermore, do the discretized and continuous models adequately capture the exponential decay of the history bias? One could use the models to simulate the rat's choices given their particular stimulus sequences and analyze that simulated data similar to what is shown in Figure 3e or f.

Authors' reply

We agree with this point and added the suggested plot as a new figure panel (Figure 6c). This was discussed under point #2 of the same Reviewer. As a reminder, the related text is (lines 550-557):

To investigate how the trials prior to $n-1$ would be expected to influence trial n choices according to the two models, we allowed both the discrete and continuous models, with τ already fitted, to predict rats' choices for the entire set of trials in their real temporal order. From the resulting series of choices, we used the GLM first employed for Figure 3f to solve for the weighting of the β coefficients (Eq. 2). In Figure 6c, it can be seen that for both models the coefficients decay with an exponential trend. Because the analysis pools trials with different preceding ITIs, the continuous and discrete models converge. For comparison, the GLM coefficient weights fitted on the observed rat data are shown in gray, carried over from Figure 3f.

Reviewer #2

Point #8

3. The author's findings add to an already substantial body of literature on adaptation and history

biases. In the introduction, the authors state that “The magnitude of this repulsive perceptual effect in relation to the elapsed time between $n-1$ and n has not yet been mapped out [...]”. I would disagree. Here are a few references which have investigated the strength of repulsive adaptation as a function of inter-stimulus interval:

- Measuring the strength of the TAE as a function elapsed time between adaptor ($n-1$) and test stimulus (n):
Magnussen, S., & Johnsen, T. (1986). Temporal aspects of spatial adaptation. A study of the tilt aftereffect. *Vision Research*, 26(4), 661–672. [https://doi.org/10.1016/0042-6989\(86\)90014-3](https://doi.org/10.1016/0042-6989(86)90014-3)
- Recovery time course from brief (<1s) adaptations to spatial contrast: Pavan, A., Marotti, R. B., & Campana, G. (2012). The temporal course of recovery from brief (sub-second) adaptations to spatial contrast. *Vision Research*, 62, 116–124. <https://doi.org/10.1016/j.visres.2012.04.001>
- Recovery time course from longer (>1s) adaptations to spatial contrast: Greenlee, M. W., Georgeson, M. A., Magnussen, S., & Harris, J. P. (1991). The time course of adaptation to spatial contrast. *Vision Research*, 31(2), 223–236. [https://doi.org/10.1016/0042-6989\(91\)90113-J](https://doi.org/10.1016/0042-6989(91)90113-J)

Authors' reply

We agree and are grateful for this review of the literature. We added the suggested references and refined the above mentioned statement which now reads (lines 54-57):

Repulsive effects have been detected^{21,24,25} for conditions in which the stimulus exposure in the preceding trial ($n-1$) is long compared to the time between trials ($n-1$ to n). In the current study, we examine the $n-1$ effects across time spans that are much longer than the single stimulus exposures.

Reviewer #2

Point #9

The influence of inter-stimulus interval has also been investigated for neural adaptation:

- In anaesthetized monkeys:

Patterson, C. A., Wissig, S. C., & Kohn, A. (2013). Distinct Effects of Brief and Prolonged Adaptation on Orientation Tuning in Primary Visual Cortex. *Journal of Neuroscience*, 33(2), 532–543. <https://doi.org/10.1523/JNEUROSCI.3345-12.2013> see their Figure 8.

- And in awake mice:

Jin, M., Beck, J. M., & Glickfeld, L. L. (2019). Neuronal Adaptation Reveals a Suboptimal Decoding of Orientation Tuned Populations in the Mouse Visual Cortex. *The Journal of Neuroscience*, 39(20), 3867–3881. <https://doi.org/10.1523/JNEUROSCI.3172-18.2019> see their Figure 1H.

All of these studies suggest an exponential decay of behavioral and neural adaptation across time. While this makes the current finding of the opposite effect perhaps even more intriguing, I think it's likely that the repulsion effect that the authors find in their task might be different in nature to the more classical stimulus adaptation investigated in the studies above. I would appreciate if the authors would take this literature into account, and discuss potential differences to these previous findings.

Authors' reply

The repulsive effect in the present work is not necessarily opposite to the phenomenon of adaptation. In adaptation, if sensory neurons process a strong stimulus ($n-1$), the response to the next stimulus (n) may be diminished. From the point of view of a downstream population receiving input from the sensory coding neurons, stimulus n will be decoded as weaker by virtue of the adapted upstream sensory coding population. This is depicted in Figure 7a under the subtitle "Sensory response is history-dependent." We have added the relevant citations to the text (lines 681-684; new text underlined):

The second form of shift calls to mind the rescaling of the sensory system coding metric according to ongoing context, a kind of adaptation believed to maximize information

transmission in a varying environment^{57,58}, where some effects have been shown to decay exponentially over time^{59,60}.

Reviewer #2

Point #10

The authors further appear to suggest that nothing is known about the temporal dynamics of repulsion across longer sequences of trials (“[...] has not yet been mapped out [...] nor has the stimulus-driven repulsion over longer sequences of trials.”). Again, I think this is not entirely correct, and here are some references that could be discussed.

- Chopin, A., & Mamassian, P. (2012). Predictive Properties of Visual Adaptation. *Current Biology*, 22(7), 622–626. <https://doi.org/10.1016/j.cub.2012.02.021>
- Gekas, N., McDermott, K. C., & Mamassian, P. (2019). Disambiguating serial effects of multiple timescales. *Journal of Vision*, 19(6), 24. <https://doi.org/10.1167/19.6.24>
- Fritsche, M., Spaak, E., & de Lange, F. P. (2020). A Bayesian and efficient observer model explains concurrent attractive and repulsive history biases in visual perception. *ELife*, 9, e55389. <https://doi.org/10.7554/eLife.55389>

Authors' reply

It was not our intention to communicate that nothing is known about the temporal dynamics of repulsion across longer sequences of trials. While the third reference was already discussed in the original manuscript, the other two citations were lacking. We have added them here in the context of the need to develop a mechanistic framework (lines 651-653):

In earlier work, it was unknown what mechanism might determine the weighting of past stimuli, especially in specifying effects as deriving from elapsed time versus the number of successive stimulus presentations^{17,53,54}

Reviewer #2

Point #11

Finally, quite recently a few papers have been published on choice repetition effects in rats and humans (e.g. Lak et al., 2020; Mendonça et al., 2020). In the context of these papers, I find it somewhat curious that the authors do not find any effects of previous choices. I do not expect the authors to offer a comprehensive explanation of these apparent differences, but I do think that these studies should be mentioned, to contextualize the current findings.

Lak, A., Hueske, E., Hirokawa, J., Masset, P., Ott, T., Urai, A. E., Donner, T. H., Carandini, M., Tonegawa, S., Uchida, N., & Kepecs, A. (2020). Reinforcement biases subsequent perceptual decisions when confidence is low, a widespread behavioral phenomenon. *ELife*, 9, e49834. <https://doi.org/10.7554/eLife.49834>

Mendonça, A. G., Drugowitsch, J., Vicente, M. I., DeWitt, E. E. J., Pouget, A., & Mainen, Z. F. (2020). The impact of learning on perceptual decisions and its implication for speed-accuracy tradeoffs. *Nature Communications*, 11(1), 2757. <https://doi.org/10.1038/s41467-020-16196-7>

Authors' reply

We agree. We added the suggested references in the Discussion sentence (lines 609-612):

While at early stages of skill learning and rule learning, choices are driven by reward contingencies³⁹⁻⁴¹, in tasks that promote faster decision-making, such as when stimulus presentation is self-terminated by the subject, rewarded choices can attract those in subsequent trials, known as “win-stay-lose-switch” (e.g.⁴²⁻⁴⁵).

Reviewer #2

Point #11

Minor comments:

4. Figure 2c: The only way in which the rat can figure out the stimulus range of the current session with certainty is by encountering a 28 mm/s or 163 mm/s stimulus, which are exclusive to the

respective stimulus ranges. Have the authors compared sessions in which neither of these two stimuli were presented during the first 3 trials? If the curves would still be separated in this comparison, this may suggest that rats did not learn the stimulus range based on the current session, but rather inferred it from the previous session's range and the alternation scheme.

Authors' reply

Good point. We agree with the reviewer's insight. Unfortunately, it would require many more sessions to build psychometric curves for the first 3 trials conditional on which stimuli were presented. Furthermore, we must point out a paradox: if certain stimuli are not presented, then the psychometric curve is, by definition, incomplete!

In reply to this Reviewer's point, as well as Reviewer #1, point #6, we have carried out new analysis, and changed the text. We tested the evolution within the session of the separation between psychometric curves by progressively adding trials, starting from the onset of the session. For each set of trials, 1 to n , we compared the individual-session PSEs across the two conditions. A significant separation ($p < 0.05$) in PSE and p ('strong') arises about 30 trials into the session. While the curves that begin to form and separate in Figure 2c are consistent with what happens later in the session, it is not quantitatively justified to make a claim. We are grateful for this clarification.

We have reinterpreted the findings now to emphasize a different, but related point: that the lack of significant separation means that the curve separation from the preceding session was not carried over. The relevant text, completely rewritten, is now (lines 165-176):

To evaluate the time course of this adjustment, Figure 2c shows the average psychometric curves of rats across the first 3 trials of every session, yielding two main findings. First, at the session onset the rats showed low performance (shallow logistic functions and high lapse rates), consistent with the results of Figure 2a. Second, the absence of significant separation between the two curves suggests that the psychometric function of the preceding session was not carried over. If it were carried over, the blue curve would have been displaced to the left of the light green curve, inasmuch as the current low-range session would initiate with a choice function from the preceding high-range session, and vice versa. Further analyses showed that the separation between the curves became statistically significant after about 30

trials (rank-sum test between PSEs, $p < 0.05$; $p < 0.001$ after about 50 trials). In conclusion, rats initiated the session with poor performance and without any observable residual influence of the previous session's range and/or boundary; their behavior adapted over time to the current session.

Reviewer #2

Point #12

5. Figure 2b and c: Is there a specific reason to show p ("strong") in the scatterplots? This metric is strongly correlated with the PSE. I would rather appreciate non-overlapping representations of the bootstrapped PSEs (inset in panel c). The distributions appear to overlap quite a bit, but this is difficult to see when the data points are plotted on top of each other.

Authors' reply

In Figure 2b, the two sets of sessions (High range and Low range) are not distinct in p ("strong") but are separated in PSE. In Figure 2c, there is strong overlap reflecting that the responses after 3 trials have, at best, only begun to incorporate the current stimulus range. We judge this presentation as providing the best means of testing the hypothesis.

Reviewer #2

Point #13

6. Figure 2e: While the curves are indeed separated, the rats show only weak adjustments to the two different stimulus distributions. When comparing Figure 2d and e, it seems that it is only the PSE in the dark condition that shifts rightwards, whereas the PSE in the light condition remains put. Could this be evidence that rats do not use the full probabilistic representation of the stimulus distribution to adjust their criterion?

Authors' reply

We agree with this point. The size of the data set encourages us to make psychometric curve comparisons for the two paired conditions – the green versus blue curves of Figure 2d and the

green versus blue curves of Figure 2e – but comparisons across conditions are not reliable. The apparently greater rightward shift of the blue curve as compared to the leftward shift of the green curve is interesting. An interesting possibility is that stimulus sp at the low end (43, 58, 73 mm/s) are in a nearly “flat” region of the sensory system’s coding range, such that increasing their probability has little effect on “pulling” the decision criterion; the responses to different very low sp values may all exert equivalent attraction. On the other hand, stimuli at high end (133, 148, 163 mm/s) might reside in a region of the sensory system’s coding range with positive slope, such that increasing their probability is more effective in “pulling” the decision criterion. This scenario would explain the elasticity in the blue curve shifting to the right from Figure 2d to e, but the green curve failing to shift to the left from Figure 2d to e. While we appreciate the Reviewer noticing these details, we do not feel confident enough about this speculation to include it in the manuscript.

Reviewer #2

Point #14

7. Perhaps it’s just me, but what the authors call “dark green” in Figure 2 looks very blue to me. Perhaps adjust the labels to avoid confusion.

Authors’ reply

We agree with this point and have changed the labels.

Reviewer #2

Point #15

8. Supplementary Figure S1a: Could the authors show a similar plot for previous incorrect choices? Relatedly, if the bias would be driven by the previous choice rather than the previous stimulus, the slopes of the green and purple curves in Figure 3 should have opposite signs, right? Perhaps, the authors could point this out to further substantiate the role of the previous stimulus rather than choice.

Authors’ reply

We have made plots for previous incorrect choices, but with rats performing at over 80% correct, the number of incorrect choices was much lower than correct, leading to a very noisy plot which we prefer not to present. It is absolutely correct that if the bias were driven by the previous choice rather than the previous stimulus, the slopes of the green and purple curves in Figure 3c would have opposite signs. We think that Fig. 3c is a more appropriate way to visualize the same phenomenon, but relying on a measure (the average rat's decision in trial n) that is more robust to the small sample size available for incorrect trials. We have added a statement to this effect (lines 260-262):

Furthermore, if the bias were driven by the previous choice rather than the previous stimulus, the violet curve would show a positive slope, opposite to that of the green curve.

Reviewer #2

Point #16

9. Figure 3c: I am not entirely convinced that it is justified to state that the bias is mostly independent of the outcome of trial $n-1$. Visually, the slopes and/or asymptotes appear to differ between previous correct and incorrect trials. In order to shed more light on this, the authors could run a model-based analysis. Specifically, the authors could fit the data with logistic functions (including "lapse" rates), quantify the difference in slopes, and compare this empirical difference with a permutation distribution of slope differences (permute the labels of previous correct/incorrect trials, re-fit logistic models, record slope difference, repeat 1000 times). Is the slope significantly steeper after previous correct trials? Are the asymptotes significantly different? This analysis would provide a more detailed view on the role of the previous outcome/reward.

Authors' reply

The same issue was raised by Reviewer #1, under point #4. Our reply is the same as above. We pursued further statistical tests and found a minor but significant effect. Specifically, the GLM

$p(\text{judged } strong)_n \sim 1 + \text{Stimulus}_n + \text{Stimulus}_{n-1} * \text{Reward}_{n-1}$

reveals that previous reward interacted weakly with stimulus history. Specifically, the sign of the coefficients reflects a decreased influence of the previous stimulus following incorrect trials. That decreased influence explains the flatter shape of the purple curve with respect to the green curve.

Reply Letter Table 2

	Estimate	SEM	p-value
Intercept	0.0299	0.0093	0.001
Stimulus_n	0.3273	0.0020	<0.001
Stimulus_n-1	-0.0245	0.0046	<0.001
Reward_n-1	-0.0189	0.0106	0.075
Stimulus_n-1:Reward_n-1	-0.0461	0.0049	<0.001

Additionally, in line with the reviewer's request, we performed a bootstrap test with the psychometric parameters of the two curves in Figure 3c following correct and incorrect trials (procedure similar to the investigation of the effect of previous choice under point #3; also see Supplementary Figure S4b). The test did not exclude a null difference in midpoint and slope (confidence level < 90%), however it evidenced a difference in the asymptote parameters. Both lapse parameters, γ and λ , seemed to decrease after an incorrect trial (confidence = 92%, 97%, respectively). Therefore, the interaction between previous stimulus and previous reward, estimated by the GLM, could be explained at a finer level as an increase in lapse probability following incorrect trials.

The interpretation is complex. The apparent attractive effect could be caused by an increased probability to lapse or to switch side following incorrect trials, as suggested by the γ and λ parameters. However, an additional factor might be at work. The sensory-perceptual process on trial n might be conditioned by a non-veridical sensory representation after an incorrect (unrewarded) trial $n-1$. If stimulus $n-1$ was weak but incorrectly judged as strong, why did the error occur? It is possible that $n-1$ was erroneously encoded by the sensory system as stronger than its mean expected value (due to noise or fluctuations); the trial n representation would then be *less repulsed from weak* as a consequence of $n-1$ being erroneously encoded as strong.

Above, we depict just two possible explanations of the small separation between curves of Figure 3c – a higher-order effect linked to lapsing or switching and a lower-order effect linked to sensory coding dynamics. Until one knows all possible causes of error, and which cause was at play on individual trials, it is difficult to hazard an interpretation of the separation between the two curves.

The modified text is the following (lines 265-272):

A logistic regression model including an interaction term between previous stimulus and previous reward was fitted to the data. The interaction weight was statistically significant ($p < 0.001$; see Supplementary Table T1), indicating that the repulsive effect of stimulus $n-1$ was slightly reduced if that trial's choice was not rewarded. Bootstrapping the parameters of the psychometric curves from Fig. 3c (see Figure S4a) revealed this phenomenon to be mainly driven by an increase in lapse probability following an incorrect trial, suggesting a slight lose-switch tendency which might partly counterbalance the effect of previous stimulus after incorrect trials.

Reviewer #2

Point #17

10. Supplementary Figure S2b: “Regression slopes for $p(\text{judged “strong”}) \times \Delta\text{speed}$ in trial $n-1$, for each trial n stimulus value.” I don't understand. What is “ $p(\text{judged “strong”}) \times \Delta\text{speed}$ ”? To look at the effect of the previous stimulus, I would expect to look at the effect of the Δspeed_{n-1} regressor.

Supplementary Figure S2c: I am not able to follow the analysis here based on the description in the legend. Is the plot in panel c a depiction of current PSE as a function of current mean of the Gaussian cdf, color coded by previous speed on correct trials? I would appreciate if the authors would elaborate on the rationale of this analysis.

Authors' reply

We agree with these points and have improved the formulations of the tests and the graphical presentation (the refined figures are now named S3b-c). We have removed “X” and stated the test as

Slopes of linear fits for $p(\text{judged “strong”})$ regressed on Δspeed in trial $n-1$, for each trial n stimulus value

This test effectively takes Δspeed_{n-1} as a regressor, as suggested.

As to Figure S2c (now named as S3c), the rationale for these analyses was to study the relation between the PSE and the psychometric parameters as a function of previous stimulus, not trial outcome. History dependent variation in PSE can be much better explained by a change in midpoint (horizontal shift) than lapse parameters (vertical shift).

Reviewer #2

Point #18

11. Figure 4d: It took me a while to understand that the two ROC curves are plotted on top of each other, since the blue area looks like a border of the grey area. It does make sense now, but I am wondering whether there would be a clearer way of showing that these are two overlapping areas and not just one grey area with a blue border.

Authors’ reply

We agree that Figure 4d was not clear and we have removed it. In its place we have included a simpler and more complete analysis of the same type; specifically, we have plotted histograms giving the difference in the area under the ROC (now labeled AUROC for better clarity) between the no-history predictor and the history model. Each AUROC value derives from one round of cross-validation. The upper histogram indicates the improved predictions for all Δspeed values on trial n , while the lower histogram focuses on the improved predictions for Δspeed of -1 on trial n , the stimulus value for which the model afforded the largest benefit. The relevant text (lines 402-409) is:

The Area Under the Receiver Operating Curve (AUROC) of the history model was 0.84 on average, while for the no-history model it was 0.83 (Figure 4d, top), an improvement of around 0.5% ($p < 0.001$, corrected resampled t-test; Cohen's $d = 4.1$). As mentioned, Brier score improvement was higher when considering current stimuli in the center of the range, peaking at $\Delta speed = -1$ (likely due to a slight imbalance in the rats' decisions; Figure 4c; see also Figure S2b). Computing the AUROC for $\Delta speed = -1$ trials (Figure 4d, bottom), the no-history and history models had values of 0.52 and 0.58, respectively (a 10% improvement; $p < 0.001$, corrected resampled t-test; Cohen's $d = 3.6$).

Reviewer #2

Point #19

12. Figure 5c and d: From the figure caption, I take that the human data was pooled and then split according to grand median ITI. This raises the possibility that differences in adaptation strength may be driven by between-subject variability in response times rather than between-trial variability. Does the regression pattern look similar when splitting ITIs per participant and then analyze the history effect based on these splits? Similarly, were the ITI quartiles for the rat data computed per rat or for the pooled data?

Authors' reply

What was written in the caption was incomplete. We did not split the data according to the grand median ITI, but according to each human subjects'/rats' median, in order to avoid confounds due to inter-subject variability. Accordingly, we have changed the caption.

Reviewer #2

Point #20

Figure 5e: It would be more informative if the datapoints would be color coded for rat identity, rather than redundantly coding the ITI. This would allow to see whether there might be systematic differences between rats (e.g. Rat 12 had quite fast ITIs overall, Rat 9 was quite slow – is the trend of the ITI repeated within each rat, or mainly driven by between-rat differences)?

Authors' reply

The Reviewer raised a similar issue under point #4. We therefore concluded that Figure 5e is not the optimal representation of the data and have eliminated it. Due to considerable rat-by-rat differences in the temporal dynamics of the stimulus history bias, the averaged dynamics as well leads to distortions rather than the addition of new information. We apologize for submitting a poorly conceived plot in the original submission.

Differences in ITI between rats do not appear to explain either their performance or their τ . The correlation between average ITI for each rat and estimated τ was also investigated in the original manuscript, leading to a non-significant result (lines 576-579):

The fact that rats' τ magnitudes correlated with their performances, while being uncorrelated with the rats' average ITI durations ($R = -0.03$, $p = 0.42$), supports the view of τ corresponding to a relevant behavioral measure, explaining a significant degree of accuracy.

Reviewer #2

Point #21

13. Do the human participants show effects of their previous choice or feedback? How does the repulsion bias decay over trials. Currently, there is only one figure panel showing the human data. While panel shows the replication of the most important finding, I think it would be nice to show that the human biases are comparable to the rat biases in other ways too (i.e. only effects of previous stimulus, not choice or reward; exponential decay across trials). This could be a supplementary figure.

Authors' reply

The rat data of tens of thousands of trials per animal supported a very detailed analysis. In humans, we set out to test the key rat result, namely, that the presentation of a weak $n-1$ stimulus

increases the likelihood of stimulus n being judged strong, and the presentation of a strong $n-1$ stimulus increases the likelihood of stimulus n being judged weak. We are able to report that phenomenon, but the data do not permit (numerically) the full array of conditional analysis done in rats.

It is worth emphasizing again why the large number of trials is required: multiple features are highly collinear. For instance, suppose one intends to isolate the effect of $n-1$ choice from the effect of $n-1$ stimulus. But $n-1$ stimulus strongly predicts $n-1$ choice. One can, conceptually, consider each $n-1$ stimulus value and separate the data into sets of trials corresponding to the two opposing choices. But due to good performance, certain stimuli ($\Delta\text{speed} = -4, -3, 3, 4$) almost always lead to the same choice. Now, if sorted into choice, one must further consider that of the two possible choices, one is correct and one is incorrect. There remains then the potential factor of correct versus incorrect. In short, the tens of thousands of trials required to sort out the individual features was not possible with humans.

As soon as covid-19 restrictions are lifted, we plan to collect a large data set from human subjects that will explore, among other things, the extent to which preceding trial influence depends on whether the subject receives feedback as to the correctness of the choice. This cannot be accomplished in rats, who work only if rewarded (e.g. informed) for correct choices. The data set was partially acquired before shut down.

Reviewer #2

Point #22

14. P. 19, line 521: “The AUC of the ROC for the continuous, time-dependent model was significantly greater than that of the discretized model (Figure 6c), demonstrating the continuous model’s more accurate predictions.”

I would not use the term “significantly” to avoid confusion, as no inferential static is provided.

Authors’ reply

We agree with this point. We have added statistical information about the comparison between

the two models to the text (lines 541-548):

When considering all trials in the dataset, AUROC was slightly but significantly higher for the continuous model compared to the discrete model (0.015% higher; $p < 0.001$, corrected resampled t-test; Cohen's $d = 0.66$; see Supplementary Figure S8). When considering exclusively the judgements for $\Delta speed = -1$, for which decisions are the most history-driven (likely due to a slight imbalance in the rats' decisions, see Figure 4c, Figure S2b, and Figure S5) the increase in AUROC was greater (about 0.32%; $p < 0.001$, corrected resampled t-test; Cohen's $d = 0.8$; see Supplementary Figure S8). Brier score was also significantly lower (corrected resampled t-test, $p < 0.001$). Together, these results indicate a higher predictive power of the continuous model with respect to the discrete model.

Reviewer #2

Point #23

15. P.26, line 712. "Contraction bias might be explained if the interaction between the short term buffer and μ is symmetric (reciprocating arrows in Figure 7b)."

It is not clear to me why the interaction would have to be symmetric. It would be helpful if the authors could elaborate on this.

Authors' reply

Thank you for this note. The chosen word, "symmetric," was not ideal. We have rephrased the sentence (lines 749-750):

Contraction bias might be explained if μ exerts an attractive force on the short term buffer, reciprocating the attraction exerted by the short term buffer on μ (opposing arrows in Figure 7b).

A further issue expressed in the Reviewer's query is whether the two reciprocating attractive forces are equal in magnitude. We spelled out our speculation on this for the same Reviewer, under point #6.

Reviewer #3

Point #1

This manuscript by Hachen et al. aimed to dissect how different aspects of trial history, such as stimulus, choice, and reward, may affect current trial decision, by training rats and humans to perform a tactile categorization task. Based on the results, the authors concluded that previous stimulus, but not choice, influences current decision, and this effect tends to intensify as inter-trial-interval prolongs. A simple model is able to capture this history-dependent behavioral effect.

This is an intriguing study. However, I have the following comments for the authors to consider and to make certain that the claims are valid conclusions of the evidence.

Authors' reply

Thank you.

Reviewer #3

Point #2

Major comments:

In Fig. 2a, to study how rats' performance evolves in a single session, the authors constructed psychometric curves from the first 5 trials in each session and compared it to a shuffled control. What is the rationale of choosing the first 5 trials, given that each session consists of several hundred trials and 9 different stimulus levels?

It appears to me that a more direct approach would be to systematically examine how psychometric curves evolve over time.

A comment of the similar nature applies to Fig. 2b and c. The observation that the relative positions of the two psychometric curves don't change when all trials are included or only the first 3 trials are included does not warrant the conclusion that the previous session exerts no effect on the performance of the current session. Additional analyses are needed to support this claim.

Authors' reply

We agree that Figure 2a does not provide a complete picture of the evolution of performance during a session. We now provide Supplementary Figure S1, showing psychometric functions fitted using trials from progressive stages of the session. For Figure 2a, we chose the first 5 trials in order to isolate the onset of the session and gain a rough estimate of the rats' initial behavior.

We agree that the data in Figure 2c do not warrant the conclusion that the previous session exerts no effect on the performance on the current session. We now limit our conclusion to stating that the psychometric curve from the previous session does not carry over into the current session. This is addressed in subsequent points of this letter.

Reviewer #3

Point #3

It also seems arbitrary to use the first 5 trials for one analysis, and the first 3 trials in another. In addition, the results in Fig. 2c seem to have large variability. To what extent are the separations of the two curves significant? Is this separation consistent across individual rats?

Authors' reply

We agree, the use of exactly 5 trials is somewhat arbitrary, yet by considering many sessions together, the first 5 trials seem like a reasonable indicator of the rats' decision curves at session onset. As mentioned above, we have extended the analysis in Supplementary Figure S1.

The reviewer is correct that the data presented in Fig. 2c do not prove a separation between the two curves. Reviewer #1 raised the same point, under point #6 and we use the same reply here. The bootstrap test between the two conditions does not exclude a null difference either for PSE or for p (‘strong’), even at the 90% confidence level.

As an additional analysis, we tested the evolution within the session of the separation between psychometric curves by progressively adding trials, starting from the onset of the session. For each set of trials, 1 to n , we compared the individual-session PSEs across the two conditions. A significant separation ($p < 0.05$) in PSE and p (‘strong’) arises about 30 trials into the session. While the curves that begin to form and separate in Figure 2c are consistent with what happens later in the session, it is not quantitatively justified to make a claim. We are grateful for this clarification.

We have reinterpreted the findings now to emphasize a different, but related point: that the lack of significant separation means that the curve separation from the preceding session was not carried over. The relevant text, completely rewritten, is now (lines 165-176):

To evaluate the time course of this adjustment, Figure 2c shows the average psychometric curves of rats across the first 3 trials of every session, yielding two main findings. First, at the session onset the rats showed low performance (shallow logistic functions and high lapse rates), consistent with the results of Figure 2a. Second, the absence of significant separation between the two curves suggests that the psychometric function of the preceding session was not carried over. If it were carried over, the blue curve would have been displaced to the left of the light green curve, inasmuch as the current low-range session would initiate with a choice function from the preceding high-range session, and vice versa. Further analyses showed that the separation between the curves became statistically significant after about 30 trials (rank-sum test between PSEs, $p < 0.05$; $p < 0.001$ after about 50 trials). In conclusion, rats initiated the session with poor performance and without any observable residual influence of the previous session’s range and/or boundary; their behavior adapted over time to the current session.

Reviewer #3

Point #4

Fig. 2d and e tried to understand how rats solve this task, setting an internal threshold or detecting stimulus distribution. The results are intriguing and confounding at the same time. The authors showed that 1. Psychometric curves don't change when shifting the stimulus boundary, and 2. Psychometric curves do change when shifting both the boundary AND the distribution. I don't think these findings warrant the conclusion that rats detect stimulus distribution but not threshold/boundary. In addition, I find it difficult to comprehend the statement that 'rats detect stimulus distribution'. What does it mean? Why does their PSE follow the mean of the distribution?

Authors' reply

The conclusion that rats do not detect the boundary *per se* is based on Figure 2d. The stimulus distributions in the light green and blue sessions were identical. Only the boundary differed – 88 mm/s in the green sessions and 118 mm/s in the blue sessions. The choice data were fully equivalent notwithstanding the two different boundaries, apart from a very small difference for one stimulus, 163 mm/s. This result offers good support for the contention that changing the boundary does not change stimulus judgment.

In Figure 2e, the same two boundaries were set – 88 mm/s in the green sessions and 118 mm/s in blue session. Now the difference in psychometric curves could be reinstated by shifting the stimulus distribution.

The Reviewer may be alluding to a highly valid point: are choices always, in any experimental or real-life condition, determined by stimulus distribution rather than by reward rule boundary? The answer is probably No. As a thought experiment, one might imagine the same set of 9 stimuli, but with the boundary positioned between 8 and 9: only the absolute highest *sp* must be classified as "strong" and the lower 8 must be classified as "weak." Essentially, the single strongest stimulus must be identified as different from the others. Would the psychometric curves of Figure 1c emerge? Probably not; it is reasonable to expect that *sp* 5, 6, perhaps 7 would be frequently (and correctly) classified as "weak." Such an experiment would show that choices can be shaped by reward rule, not only by stimulus distribution. In light of this possibility, we have now modified the

text to emphasize that the psychometric curve dynamics that we observe may be specific to our experimental conditions. Beginning with line #202, we have added the new (underlined) text:

The results suggest that choices were dictated by stimulus distribution but not by boundary – the alignment of the rat’s decision criterion to the session’s category boundary, evident both in Figure 2b and in Figure 2e, seems to be an outcome of the stimulus distribution to which the rat was exposed, not the reward rule boundary or the frequency of reward associated with a given category. There likely exist experimental and real-life conditions in which perceptual choices are determined as much by reward rule as by stimulus distribution; however, the present conditions seem to cause rats to link their decisions to the statistical structure of the stimulus set. Such stimulus-dependent decision-making motivates a detailed analyses of how past stimulus values generate current choices.

We agree that the subtitle **Do rats detect the boundary or the stimulus distribution?** was inexact. We have changed the subtitle to **“Are psychometric curves modulated by the boundary or by the stimulus distribution?”**

Finally, “Why does their PSE follow the mean of the distribution?” This is compatible with (and we would argue, accounted for by) the discrete and continuous models, where the decision criterion (which is reflected in the PSE) will fluctuate around the mean of the distribution (see Figure 6a).

Reviewer #3

Point #5

Fig. 3 attempted to dissect the effects of previous stimulus amplitude and choice on current decision. The authors concluded from 3b that previous choice does not affect current decision. However, 3b only considered rewarded choices. It is not clear to me why unrewarded choices were excluded from this analysis. It is also unclear to me why 3b only used 0 previous speed and 3c used all previous speeds. If choice/reward history does not affect current decision, how did these rats learn to perform the task?

Authors’ reply

Reasons for excluding incorrect $n-1$ trials was that rewarded choices might have a different effect on the upcoming choice than unrewarded ones (rats might have a tendency to switch choice side after an unrewarded trial; see reply to Review #1, point #4). To avoid mixing rewarded and unrewarded $n-1$ trials, we excluded trials after unrewarded $n-1$ choices, thus eliminating only about 20% of trials.

We selected only trials in which trial $n-1$ $\Delta speed$ was 0 in order to eliminate any possible influence of previous stimulus' speed. This isolated trial $n-1$ choice as a factor, which was the purpose of the panel.

The purpose of Figure 3c was complementary, namely, to isolate the stimulus effect with respect to previous reward (green and purple lines). This figure gives the probability of categorizing stimulus n as "strong" as a function of the $\Delta speed$ of trial $n-1$. All $\Delta speed$ values of trial n are pooled, making $n-1$ correct (green) versus $n-1$ incorrect (violet) the critical factor. With this conditional analysis, only possible thanks to a very large data set, we were able to isolate possible effects of choice and reward specifically, reducing influences from other stimulus history features.

We have included new analyses related to Figure 3c, as described in reply to **Reviewer #1, Point #4**. We pursued further statistical tests and found a minor but significant effect of previous reward. Specifically, we formulated the GLM:

$$p(\text{judged strong})_n \sim 1 + \text{Stimulus}_n + \text{Stimulus}_{n-1} * \text{Reward}_{n-1}$$

which reveals that previous reward interacted weakly with stimulus history. Specifically, the sign of the coefficients reflects a decreased influence of the previous stimulus following incorrect trials. That decreased influence explains the flatter shape of the purple curve with respect to the green curve.

Reply Letter Table 2

	Estimate	SEM	p-value
--	----------	-----	---------

Intercept	0.0299	0.0093	0.001
Stimulus_n	0.3273	0.0020	<0.001
Stimulus_n-1	-0.0245	0.0046	<0.001
Reward_n-1	-0.0189	0.0106	0.075
Stimulus_n-1:Reward_n-1	-0.0461	0.0049	<0.001

Additionally, we performed a bootstrap test with the psychometric parameters of the two curves in Figure 3c following correct and incorrect trials (procedure similar to the investigation of the effect of previous choice under point #3; also see supplementary Figure S4b). The test did not exclude a null difference in midpoint and slope (confidence level < 90%), however it evidenced a difference in the asymptote parameters. Both lapse parameters, γ and λ , seemed to decrease after an incorrect trial (confidence = 92%, 97%, respectively). Therefore, the interaction between previous stimulus and previous reward, estimated by the GLM, could be explained at a finer level as an increase in lapse probability following incorrect trials.

The modified text is the following (lines 265-272):

A logistic regression model including an interaction term between previous stimulus and previous reward was fitted to the data. The interaction weight was statistically significant ($p < 0.001$; see Supplementary Table T1), indicating that the repulsive effect of stimulus $n-1$ was slightly reduced if that trial's choice was not rewarded. Bootstrapping the parameters of the psychometric curves from Fig. 3c (see Figure S4a) revealed this phenomenon to be mainly driven by an increase in lapse probability following an incorrect trial, suggesting a slight lose-switch tendency which might partly counterbalance the effect of previous stimulus after incorrect trials.

The Reviewer asks "If choice/reward history does not affect current decision, how did these rats learn to perform the task?" This is a fascinating question, but beyond the scope of the paper. Our speculation (restricted to the reply letter) is based on classical learning theory, namely, that learning occurs by assessment of prediction errors. During the training procedure, rats would certainly need to incorporate knowledge gained through the stimulus/choice/reward contingency. However, in the well-trained rats, the successful collection of the reward is fully predicted, and

trials without reward are attributed to processing noise and are thus do not constitute “surprises” to be learned from. In short, we think that highly trained rats are no longer “learning” in the strict sense of the term.

Reviewer #3

Point #6

The authors showed that 1. stimulus distribution determines rat choices, 2. rat behavior is different in the first 3 trials using high and low ranges, and 3. the optimal trial history is 1-3 trials for the model. So, how are the first 1-3 trials enough for the rats to establish their internal reference and correctly perform the task?

Authors' reply

We agree with the reviewer. The bootstrap test between the two conditions does not exclude a null difference either for PSE or for $p(\text{'strong'})$, even at the 90% confidence level. We no longer make the claim that the rats adjust their boundary within 3 trials. Reviewer #1 raised the same issue and we give the same reply here.

We tested the evolution within the session of the separation between psychometric curves by progressively adding trials, starting from the onset of the session. For each set of trials, 1 to n , we compared the individual-session PSEs across the two conditions. A significant separation ($p < 0.05$) in PSE and $p(\text{'strong'})$ arises about 30 trials into the session. While the curves that begin to form and separate in Figure 2c are consistent with what happens later in the session, it is not quantitatively justified to make a claim. We are grateful for this clarification.

We have reinterpreted the findings now to emphasize a different, but related point: that the lack of significant separation means that the curve separation from the preceding session was not carried over. The relevant text, completely rewritten, is now (lines 165-176):

To evaluate the time course of this adjustment, Figure 2c shows the average psychometric curves of rats across the first 3 trials of every session, yielding two main findings. First, at the session onset the rats showed low performance (shallow logistic functions and high lapse rates), consistent with the results of Figure 2a. Second, the absence of significant separation

between the two curves suggests that the psychometric function of the preceding session was not carried over. If it were carried over, the blue curve would have been displaced to the left of the light green curve, inasmuch as the current low-range session would initiate with a choice function from the preceding high-range session, and vice versa. Further analyses showed that the separation between the curves became statistically significant after about 30 trials (rank-sum test between PSEs, $p < 0.05$; $p < 0.001$ after about 50 trials). In conclusion, rats initiated the session with poor performance and without any observable residual influence of the previous session's range and/or boundary; their behavior adapted over time to the current session.

Reviewer #3

Point #7

Fig. 5 shows that longer ITI exerts stronger stimulus effects on decision. This is a bit surprising as we'd normally think such effects decay over time. I wonder whether there are other factors that may underlie this effect. For example, are ITIs correlated with task difficulty? Since trials are self-paced, are ITIs associated with the motivational states of the rats? The authors mentioned that they excluded the longest 20% of ITIs as they were associated with task unrelated behaviors. Are the remaining 80% completely free of such behavior?

Authors' reply

The finding that stimulus $n-1$ exerts a stronger influence on trial n choice after a longer ITI was surprising to the authors as well, and led us to formulate the model as a possible explanation.

The reason for removing such a large portion of the dataset (20%) was precisely to make sure that such task-unrelated behaviors, that we observe only in a very minor portion of the trials (e.g. 10-20 trials per session), would be completely removed. After some trials, the rat stops to groom or to explore the box, and we suspected that stimulus history might act differently in those cases.

To further confirm the absence of a confound related to slow fluctuations in ITI, we checked ITI duration as a function of trial number. This is reported below.

As to which factors determine the ITI, we agree with the reviewer's intuition that there exist many factors that correlate with both ITI and task difficulty. This issue was raised by Reviewer #2 under Point #2 and we use the same reply here. Unfortunately, there is no evidence in our study or in any study known to us as to what causes variability in ITI; it may reflect attention, but we have no means of directly monitoring attention. However, we have now checked whether ITIs shift systematically and found that they are quite stable. Furthermore, the effect of ITI on trial n choice as for the full session is also quite stable. The new analyses are consistent with the history effect on decision criterion not depending on the behavioral state, as outlined in lines 444-457:

We performed control analyses excluding potential confounds (see Supplementary Figure S6). First, we replicated Figures 5b and 5c after excluding previous incorrect trials (Figure S6). Furthermore, using only the first 150 trials of each session (when motivation or urgency might be stronger), the ITI had the same effect on trial n choice as for the full session. There was a difference of just 50 ms in median ITI for the first 150 trials of each session compared to the whole session (7.57 s versus 7.62 s), suggesting that the pace of the task was stable within sessions. [...]

Finally, a GLM analysis (Supplementary Table T2) uncovered a significant repulsive effect of stimulus $n-2$, which was reduced with increasing ITI from stimulus $n-1$ to n . This is explained by the memory of $n-2$ being "written over" as $n-1$ exerts a progressively stronger effect. Accordingly, with reduced ITI from stimulus $n-1$ to n the effect of $n-2$ was larger. Taken together, these tests argue against the possibility that slow fluctuations in ITI during the course of the session co-varied with (and might partially account for) the trial-by-trial stimulus history effect.

We also checked whether ITI duration was correlated with stimulus difficulty and found no relationship (Figure S6c); this is now reported in lines 450-451:

The difficulty of trial $n-1$ (i.e. the closeness of $\Delta speed$ to the category boundary) did not affect the ITI from $n-1$ to n (Figure S6c).

Reviewer #3

Point #8

Finally, a major missing piece in this work is an effort to reveal the neural mechanisms underlying the described behavioral effects, such as stimulus dependency and ITI dependency. Uncovering some aspects of the neuronal correlates would significantly enhance the readers' appreciation of the behavioral results and contribute to our understanding of history dependent perceptual judgement.

Authors' reply

While in ongoing and future studies we aim to uncover possible neuronal correlates underlying the behavioral findings and the basic models developed here, a systematic and fundamental search for neuronal correlates by far exceeds the scope of the current manuscript. Figure 7b shows the basic components of the neural mechanisms of reference memory, at least as posited by our model. In current research, we are trying to identify each of these components (τ , sensory representation, short term buffer) and demonstrate their roles causally through optogenetics. Though the experiments are progressing well, it will take years to complete the data set and we think that the purely behavioral data, with a computationally robust model behind them, along with human confirmation of the main phenomenon, will be of interest to the community even as we work to reveal the neural mechanisms.

Reviewer #3

Point #9

Minor comments:

Fig. 1b is confusing. The slope at inflection point is plotted in gray, and the Gaussian is also gray. Gray Y-axis is not labeled and it's unclear whether it is designating y-values for the Gaussian or the inflection point or both.

Authors' reply

Thank you for pointing out these details. We have improved the figure.

Reviewer #3

Point #10

In a number of figure panels, axis values are not aligned to tick marks, reducing readability.

Authors' reply

We carefully checked the quality of the figures and improved the alignments.

Reviewer #3

Point #11

In Fig. 2e, why PSE = 118 corresponds to a 40% performance, if the definition of PSE is 50% performance?

Authors' reply

There is a misunderstanding here; the $p(\text{'strong'})$ value on the y axis is not the y-value corresponding to the PSE, which is in fact 0.5, but it is the average $p(\text{'strong'})$ for the entire function. In other words, it is the overall probability of choices to the strong category, over the entire stimulus set. Roughly speaking, it is proportional to the area under the psychometric curve.

Reviewer #3

Point #12

Evidence detailed in Fig. 3 e and f suggests trial history is strongest for $n-1$, but the discretized model used $\tau = 3.3$ trials for the trial history. Some discussion regarding this difference is needed.

Authors' reply

The τ parameter does not indicate which of the past trials that exerts the strongest effect on the

current trial (in fact this is always $n-1$, the immediately preceding trial); instead, τ corresponds to the point where 63.2% of the effect the trial will have decayed, which indeed happens at around trial $n-3$ in both Fig. 3e and f.

Reviewer #3

Point #13

Line 627-628: It is suggested that no terms are required for more distant trials other than $n-1$. What is the justification? No evidence was shown for the updated model that suggests trial $n-1$ is the optimal number of trial history. The discretized model used $\tau = 3.3$ trial, for example.

Authors' reply

Both models generate the decision boundary as a sliding average with exponential decay. Thereby all previous trials exert an effect on the choice in trial n , however this effect decays exponentially in the more distant past. In this way the strongest effect is created by the previous trial, but the effect acts upon the previously generated decision boundary and so on. In the Rescorla-Wagner model, corresponding dynamics of behavioral biases are described, where in order to keep a fixed learning rate from trial to trial, the memory of previous trials decays exponentially. As mentioned above, the value of $\tau = 3.3$ should not be considered as the optimal trial, but rather the distance in the past at which, on average, a trial's weight will have decayed by 63%.

Reviewer #3

Point #14

The models discussed in this paper did not include noise as a factor. Biological systems are subject to both intrinsic and extrinsic noise. Although not necessarily detrimental to the study, including noise (either for the sensory memory trace or the decision criterion) may improve model performance.

Authors' reply

This is an interesting point. In our work, the noise is captured by the cumulative Gaussian function, depicted in Figure 1b. This incorporates trial-to-trial variability, making each sensory response a probabilistic curve.

Beyond that, why are representations in the brain “noisy”? One form of noise is that when the same physical stimulus is presented on many instances, the brain encodes the stimulus and/or drives a decision according not only the sensory data in the receptor but also according to the history of preceding trials. Thus, one can consider the present study as an attempt to sort out one source of noise, history-dependent intrinsic noise.

Reviewer #3

Point #15

Fig. 6c: Is the Y axis miss labelled? Otherwise, is it meaningful to compare 2 measures when their AUC difference is on the order of e^{-15} ? In addition, are the predictions in 6c an average of all speeds? It is important to compare model predictions in the 0 speed condition.

Authors' reply

We have moved the AUC (now labeled AUROC for better clarity) plots to the Supplementary sections. A reworked figure that carries the same message as the previous Figure 6c is now shown in Supplementary Figure S8. We are grateful to the Reviewer for detecting a mistake in the original Figure 6c plot. The average AUC difference is larger by several orders of magnitude ($\sim 1e^{-4}$), and the effect size is also large (Cohen's $d = 0.66$; $p < 0.001$).

In agreement with the reviewer's wish, we also focused on the stimulus in which the discrete model performed best (i.e. for which decisions are the most history-driven); this turned out to be $\Delta\text{speed} = -1$, likely due to a slight imbalance in the rats' decisions (see Figure 4c, Figure S2b, and Figure S5). Restricted to this stimulus, the increase in AUROC is much larger ($\sim 2e^{-3}$, Cohen's $d = 0.8$; $p < 0.001$; see Figure S8b).

The relevant text is (lines 541-548):

When considering all trials in the dataset, AUROC was slightly but significantly higher for the continuous model compared to the discrete model (0.015% higher; $p < 0.001$, corrected resampled t-test; Cohen's $d = 0.66$; see Supplementary Figure S8). When considering exclusively the judgements for $\Delta speed = -1$, for which decisions are the most history-driven (likely due to a slight imbalance in the rats' decisions, see Figure 4c, Figure S2b, and Figure S5) the increase in AUROC was greater (about 0.32%; $p < 0.001$, corrected resampled t-test; Cohen's $d = 0.8$; see Supplementary Figure S8). Brier score was also significantly lower (corrected resampled t-test, $p < 0.001$). Together, these results indicate a higher predictive power of the continuous model with respect to the discrete model.

Figure 6c has been replaced by a plot requested by all Reviewers which shows that both models, discrete and continuous, predict the same exponential decay in the weight of preceding stimuli as seen in the actual observations.

Reviewer #3

Point #16

Major stylistic revisions are needed to enhance readability.

Authors' reply

According to the helpful comments and suggestions of all the three Reviewers we improved the readability of the manuscript.

Reviewers' Comments:

Reviewer #1:

Remarks to the Author:

The authors have comprehensively addressed my concerns, including some interesting new analyses that further strengthen the study.

Reviewer #2:

Remarks to the Author:

I would like to thank the authors for their thoughtful responses to my comments. Most comments have been addressed satisfactorily. There are only a few minor points remaining. Overall, I think this study will be an important addition to the literature.

1. Reply to my comment R2, point 21: The authors argue that they do not have enough data from the human experiment to investigate the influence of previous choice and reward on current choice. However, if I understand correctly, they regress out the effect of previous choice and only consider previous correct trials when presenting the human data (Fig. 5d; described in the figure caption). This suggests that there are quantifiable effects of previous choice and reward (correct vs. incorrect) – why else would one need to regress out their effects? How would Fig. 5d look when not regressing out the effects of previous choice and considering both correct and incorrect previous trials? I believe that providing more details on this is important for the following reason. It has been previously observed that humans exhibit biases to repeat their previous choice (attraction bias) and this attraction bias is increased for previous choices with fast response times (Urai et al., 2017; Bosch et al., 2020). If such an effect is also present in the current human data, and is not properly accounted for, it could contribute to the pattern of decreased stimulus repulsion after fast ITIs. That is, an increase in choice attraction after short ITIs (due to short RTs) could lead to an apparent decrease in stimulus repulsion after short ITIs, when these two effects are superimposed. The careful analysis of the rat data convinced me that this cannot explain the effect of ITI in the rat data. It would be important to show that the same holds for the human data too. The fact that the authors regress out choice biases in the human data makes me somewhat wary about this point. I concur with the authors' argument that the presentation of the human data should serve to test the key rat result, and is not meant to provide a thorough investigation of human history biases. However, it would be important to ensure that the rat and human key results are truly consistent, not just bearing a superficial resemblance.

Urai, A. E., Braun, A. & Donner, T. H. Pupil-linked arousal is driven by decision uncertainty and alters serial choice bias. *Nat Commun* 8, 14637 (2017).

Bosch, E., Fritsche, M., Ehinger, B. V. & de Lange, F. P. Opposite effects of choice history and evidence history resolve a paradox of sequential choice bias. *Journal of Vision* 20, 9 (2020).

2. One small stylistic suggestion for line 175: The authors could clarify that the significant separation between the curves is following adjustments to the *current* session's range, not that of the previous session. In the preceding sentence, they write about the absence of a significant separation, but this pertains to the adjustment to the previous session. Readers might become confused and think both sentences refer to the same separation.

3. The references to the supplemental figures are not properly updated in the manuscript. For instance:

Line 414 and 553: Figure S2b -> Figure S3b

Line 467: Supplementary Figure S5 -> Supplementary Figure S7

There might be more instances.

4. The reference list is not complete. Some references from the rebuttal are not in the list, and some numbers in the text point to wrong references in the list.

Reviewer #3:

Remarks to the Author:

We appreciate the efforts the authors took to address reviewers' comments. We still have the following questions, some of which are remaining questions and not addressed, and some are new.

Response to point #3: 1. Our question about the inconsistency in using first 3 and 5 trials is not addressed. 2. A graphic illustration for the statement 'Further analyses showed that the separation between the curves became statistically significant after about 30 trials (rank-sum test between PSEs, $p < 0.05$; $p < 0.001$ after about 50 trials)' would be useful.

Response to point #4: 1. Even though the curves are separated, the low stimulus distribution PSE is still the same for both Fig. 2d and 2e. If the PSE is distribution dependent, are they supposed to be different? 2. To better address this question, we believe additional experiments are needed where the boundary is held constant while the distribution changes.

Response to point #5: 1. The authors' reasoning to exclude the unrewarded trials is what we were asking in the first place, i.e., whether choice or reward history affects current decision. Since psychometric curves can be constructed based on 3 or 5 trials, 20% incorrect trials should be sufficient to draw conclusions. 2. In addition, what is the accuracy of the GLM included in the response, i.e., to what extent can the GLM predict behavioral outcome. 3. The last sentence of the response 'In short, we think that highly trained rats are no longer "learning" in the strict sense of the term', possibly contradict some earlier results: 1) Well-trained rats improve their performance over time with each session (Fig. 2), and 2) Some of them can learn a variant of the task (Fig. 2).

Response to point #10: Fig. S4a, x axis in the right two panels seems incomplete? Fig. S5a, b, no labels for the y axis in the right panels?

Reviewer #1 (Remarks to the Author):

The authors have comprehensively addressed my concerns, including some interesting new analyses that further strengthen the study.

Authors' Reply

Thank you.

Reviewer #2 (Remarks to the Author):

I would like to thank the authors for their thoughtful responses to my comments. Most comments have been addressed satisfactorily. There are only a few minor points remaining. Overall, I think this study will be an important addition to the literature.

1. Reply to my comment R2, point 21: The authors argue that they do not have enough data from the human experiment to investigate the influence of previous choice and reward on current choice. However, if I understand correctly, they regress out the effect of previous choice and only consider previous correct trials when presenting the human data (Fig. 5d; described in the figure caption). This suggests that there are quantifiable effects of previous choice and reward (correct vs. incorrect) – why else would one need to regress out their effects? How would Fig. 5d look when not regressing out the effects of previous choice and considering both correct and incorrect previous trials? I believe that providing more details on this is important for the following reason.

It has been previously observed that humans exhibit biases to repeat their previous choice (attraction bias) and this attraction bias is increased for previous choices with fast response times (Urai et al., 2017; Bosch et al., 2020). If such an effect is also present in the current human data, and is not properly accounted for, it could contribute to the pattern of decreased stimulus repulsion after fast ITIs. That is, an increase in choice attraction after short ITIs (due to short RTs) could lead to an apparent decrease in stimulus repulsion after short ITIs, when these two effects are superimposed.

The careful analysis of the rat data convinced me that this cannot explain the effect of ITI in the rat data. It would be important to show that the same holds for the human data too. The fact that the authors regress out choice biases in the human data makes me somewhat wary about this point. I concur with the authors' argument that the presentation of the human data should serve to test the key rat result, and is not meant to provide a thorough investigation of human history biases. However, it would be important to ensure that the rat and human key results are truly consistent, not just bearing a superficial resemblance.

Urai, A. E., Braun, A. & Donner, T. H. Pupil-linked arousal is driven by decision uncertainty and alters serial choice bias. *Nat Commun* 8, 14637 (2017).

Bosch, E., Fritsche, M., Ehinger, B. V. & de Lange, F. P. Opposite effects of choice history and evidence history resolve a paradox of sequential choice bias. *Journal of Vision* 20, 9 (2020).

Authors' Reply

The reviewer is correct in reading that in the original submission and in the first revision, in the data of human participants we measured the bias driven by the previous stimulus only after residualizing the effect of previous choice.

We share the Reviewer's interpretations of the literature on the attractive effect of choice in humans and have created a new presentation of our analysis to explore the effect; in the new revision we find that the choice-attractive effect overlaps with (and opposes) the repulsive effect of stimulus alone.

We have now referred to the attractive choice history effect found in humans in the main text (lines 466-469):

In contrast to rats, in humans we also found a prominent attractive choice effect (in line with previous findings^{33,34}). For the purpose of isolating the stimulus history effect and its dynamics we factored out the choice effect (see Supplementary Figure S9).

We have retained Figure 5d, which plots for human subjects the slope of the regression line fit between bias (measured as the point of subjective equality, PSE) and previous trial ($n-1$) $\Delta speed$, separated by inter-trial interval (ITI). We have provided a reorganized Supplementary figure (S9) to better illustrate the procedure used to generate the plot of Figure 5d. In what follows, we unpack the methodology of the original and the new illustrations.

Because of the high correlation between stimulus and decision variables, a standard linear regression including $\Delta speed$ $n-1$ and choice $n-1$ would be misleading (as discussed in the previous reply letter). As a more rigorous alternative method, we proceeded as follows:

(1) In cases where trial $n-1$ $\Delta speed$ was 0 and the subject's choice was “correct,” we generated two separate trial n psychometric curves according to the choice on trial $n-1$ (“weak” or “strong”). Since the influence of stimulus $\Delta speed$ and reward on trial $n-1$ are both factored out (trial $n-1$ $\Delta speed$ being 0, and positive feedback being delivered in all instances), the two trial n psychometric curves differ only in relation to the trial $n-1$ choice.

(2) For those two trial n psychometric curves we measured the difference in PSE. We have seen in Figure 3b that rats’ trial n psychometric curves were only minimally affected by trial $n-1$ choice (small red and blue squares in Figure 3b). For humans, consistent with the literature cited by the Reviewer, this was not the case. This is now illustrated in Figure S9a upper panel, which gives PSE on trial n according to trial $n-1$ choice. The ITI’s (from $n-1$ to n) below the median (“Short ITIs”) are depicted in light blue while those above the median (“Long ITIs”) are depicted in dark blue. The plot reveals that when $n-1$ choice was “strong,” the trial n PSE was negative, indicating a tendency to choose “strong” again. This confirms the attractive sequential effect of choice, as outlined by the Reviewer. Additional observations: interestingly, the attractive effect of trial $n-1$ choice appears to be enhanced after ITIs above the median. While this is a novel and surprising result (the literature reports that attraction is weaker, not stronger, after a long ITI: Urai et al., 2017; Bosch et al., 2020), we do not pursue this observation further at present because earlier reports refer to ITIs of up to a

few seconds at most, while our study includes ITIs of 6 to 12 s, so the comparison would be ill-posed. In the future, we will generate a larger human data set to examine the dynamics of both stimulus-repulsion (extending the current work) and choice-attraction.

(3) As shown in S9a middle panel, we then measured the PSE in trial n as a function of $\Delta speed$ $n-1$, separated by long and short ITI, but without factoring out the $n-1$ choice effect (labeled “before residualization”). This is comparable to Figure 5c for rat data. In this plot, a positive slope is evident in the set of points where $n-1$ $\Delta speed$ was negative and a positive slope is evident in the set of points where $n-1$ $\Delta speed$ was positive, but there is a reset at the midpoint, where $n-1$ $\Delta speed$ was 0. The two positive-sloped segments, with a reset between them, occur (we hypothesize) because of the attractive effect of $n-1$ choice, where choice itself is correlated with $\Delta speed$ of $n-1$.

(4) To isolate the effect of $\Delta speed$ of $n-1$, we “residualized” the effect of trial $n-1$ choice. Specifically, for every PSE obtained in step 3 we subtracted the effect of previous choice (measured in the upper panel a) from the effect of previous stimulus. The result is given in S9a lower panel, which also serves as main text Figure 5d (where regression fits are included).

Human participants received feedback as to whether choice was correct or incorrect, and were paid according to the total number of correct trials, so we can take the single-trial feedback as comparable to reward for the rats. Thus, the Reviewer also asks why and how the **reward** was residualized in the human studies. To answer this, we carried out a new analysis, in parallel with the above procedure:

(1) In cases where trial $n-1$ $\Delta speed$ was 0, we generated two separate trial n psychometric curves according to the choice on trial $n-1$ (“weak” or “strong”), using only **incorrect** $n-1$ trials.

(2) Again, as a measure of the effect of preceding decision, we took the difference in psychometric curve PSE on trial n , according to trial $n-1$ choice. Interestingly, when trial $n-1$ was incorrect, the associated choice no longer affected the PSE on trial n (S9b upper panel). The attractive effect of trial $n-1$ choice on trial n choice appears to depend on trial $n-1$ being “rewarded.” Alternatively, “incorrect” feedback may be interpreted by subjects as punishment. At present we cannot distinguish whether it is absence of reward or presence of punishment that cancels the choice-attractive effect. In work that is already underway, we are exploring this by running sessions with human participants with and without feedback.

(3) The absence of attractive influence of incorrect trial $n-1$ choice is seen both for short and for long ITIs. The attractive effect is simply absent and thus shows no temporal dynamics.

(4) As shown in S9b middle panel, we then measured the PSE in trial n as a function of stimulus $n-1$, separated by long and short ITI, but without factoring out the $n-1$ choice effect (labeled “before residualization”). In this plot, a positive slope is evident even before residualization, inasmuch as the attractive force of $n-1$ choice was absent and did not need to be factored out. However, because the number of trials is low, the data are noisy.

(5) To follow the same procedure as that of correct $n-1$ choices, we nonetheless residualized the effect of trial $n-1$ choice. Because incorrect $n-1$ choices had no attractive effect, the residualization procedure led to no detectable change (S9b lower panel). The repulsive effect of trial $n-1$ $\Delta speed$ remains evident.

In S9c, data following both correct and incorrect $n-1$ choices are pooled. In this case, we balanced the proportion of corresponding “strong” and “weak” $n-1$ choices, taking into account the frequency of each choice for each value of $n-1$ $\Delta speed$.

In the caption of Supplementary Figure S9, we explain the above analyses in detail.

Going back to the reviewer’s query, we shared the concern that a decrease in choice attraction with longer ITI could lead to an apparent increase in stimulus repulsion, and investigated this aspect. However, given the above analyses, the scenario is not plausible in this dataset since the choice-attractive effect was stable across ITI or, if anything, showed a trend towards getting stronger.

In sum, before accounting for the $n-1$ choice-attractive effect, the repulsive effect of $\Delta speed$ of $n-1$ is almost impossible to discern after correct choices. This is because preceding stimulus and preceding choice exert opposite forces. We understand that the choice history effect is an important issue, and we are grateful to the Reviewer for emphasizing this. In reply, we have gone through the procedure step by step (above) and have made the two history effects explicit. This problem is now one focus of the laboratory.

We think that the mechanisms described in Urai et al. and Bosch et al. are valid also for rodents. In another study, in preparation, we have been able to isolate both attractive choice effects and the repulsive stimulus history effect described here, by employing slightly different experimental protocols in rats.

Reviewer #2

2. One small stylistic suggestion for line 175: The authors could clarify that the significant separation between the curves is following adjustments to the *current* session's range, not that of the previous session. In the preceding sentence, they write about the absence of a significant separation, but this pertains to the adjustment to the previous session. Readers might become confused and think both sentences refer to the same separation.

Authors' Reply

Thank you. We have clarified the text (lines 164-175):

To evaluate the time course of this adjustment, Figure 2c shows the average psychometric curves of rats across the first 3 trials of every session, yielding two main findings. First, at the session onset the rats showed low performance (shallow logistic functions and high lapse rates), consistent with the results of Figure 2a. Second, the absence of significant separation between the two curves suggests that the psychometric function of the preceding session was not carried over to bias choices in the new session. If it were carried over, the blue curve would have been displaced to the left of the light green curve, inasmuch as the current low-range session would initiate with a choice function from the preceding high-range session, and vice versa. Further analyses (Supplementary Figure S1b) showed that the separation between the curves became statistically significant after about 30 trials (rank-sum test between PSEs, $p < 0.05$; after about 50 trials, $p < 0.001$). In conclusion, rats initiated the session with poor performance and without any observable residual influence of the previous session's range and/or boundary; their behavior adapted over time to the current session.

Reviewer #2

3. The references to the supplemental figures are not properly updated in the manuscript.

For instance:

Line 414 and 553: Figure S2b -> Figure S3b

Line 467: Supplementary Figure S5 -> Supplementary Figure S7

There might be more instances.

Authors' Reply

We have fixed these errors.

Reviewer #2

4. The reference list is not complete. Some references from the rebuttal are not in the list, and some numbers in the text point to wrong references in the list.

Authors' Reply

We have fixed these errors.

Reviewer #3 (Remarks to the Author):

Reviewer #3 (Remarks to the Author):

We appreciate the efforts the authors took to address reviewers' comments. We still have the following questions, some of which are remaining questions and not addressed, and some are new.

Response to point #3: 1. Our question about the inconsistency in using first 3 and 5 trials is not addressed.

Authors' Reply

We have changed Figure 2a; it is now based on the first 3 trials, which did not change the conclusions.

Reviewer #3

2. A graphic illustration for the statement 'Further analyses showed that the separation between the curves became statistically significant after about 30 trials (rank-sum test between PSEs, $p < 0.05$; $p < 0.001$ after about 50 trials)' would be useful.

Authors' Reply

We have now illustrated the data on which the statement is based; this is Figure S1b.

Reviewer #3

Response to point #4: 1. Even though the curves are separated, the low stimulus distribution PSE is still the same for both Fig. 2d and 2e. If the PSE is distribution dependent, are they supposed to be different? 2. To better address this question, we believe additional experiments are needed where the boundary is held constant while the distribution changes.

Authors' Reply

Two partially different sets of rats furnished the data in Figures 2d versus 2e, and there were different numbers of sessions in each test condition. One always finds somewhat different psychometric curves for different rats, and for this reason we argue that the more meaningful comparison is between the

two curves plotted within Figure 2d and then between the two curves plotted within Figure 2e. These are within-rat-group comparisons and so are more reliable. We have now added a clarification on lines 196-197: “Two rats were tested in each paradigm; 1 rat participated in both paradigms and 2 rats participated in just 1 paradigm.”

Nonetheless, we appreciate the issue raised by the Reviewer and have carried out further analysis. The figure pasted in below shows the bootstrapped PSEs arising from the 4 different experimental conditions, as in the insets of Figure 2 d-e. The two black histograms give the distribution of PSEs corresponding to Figure 2d, where the stimulus ranges and extraction probabilities were equal under the two conditions, and only the category boundary was switched between 88 and 118 mm/s. The bootstrapped PSEs are fully overlapping, indicating that the category boundary change, if not accompanied by a change in stimulus distribution, did not shift the corresponding psychometric curves.

The two red histograms give the distribution of PSEs corresponding to Figure 2e, where the category boundary was switched between 88 and 118 mm/s. While the stimulus ranges were equal, the extraction probability was held equal for the two categories, leading to a change in the extraction probability for the different stimulus values. There was no overlap between the two distributions of bootstrapped PSEs, indicating that the category boundary change did indeed lead to a shift in the corresponding psychometric curves, if accompanied by a change in stimulus distribution.

The Reviewer emphasizes the need to compare the left red histogram (category boundary at 88 mm/s, with stimulus extraction probability equal for the two categories) to the black histograms (two different category boundaries, but uniform stimulus extraction probability across all 9 stimuli). The red histogram appears left-shifted with respect to the black histograms. Although we emphasize again that the experimental design was not optimized for this specific comparison, a K-S test finds that the distributions are significantly different (Kolmogorov-Smirnov test, $p < 0.001$). In other words, it is extremely unlikely that these two observed distributions represent two sets of samples taken from the same underlying source.

Reviewer #3

Response to point #5: 1. The authors' reasoning to exclude the unrewarded trials is what we were asking in the first place, i.e., whether choice or reward history affects current decision. Since psychometric curves can be constructed based on 3 or 5 trials, 20% incorrect trials should be sufficient to draw conclusions.

Authors' Reply

The psychometric curves based on the first 5 trials (now first 3 trials) were not conditional on preceding stimulus. When separated according to the preceding stimulus, the data *per curve* is reduced by a factor of 9. The trial n psychometric curves, after the 20% incorrect trial $n-1$ choices, separated according to $\Delta speed$ of $n-1$, do become sparse and less reliable for capturing the graded

effect of previous stimulus. Having said that, we nonetheless plotted the results below, with the finding that there remains a significant, although weaker effect, of stimulus $n-1$. The p-value of the regression line here shown is 0.015. This has now been added as Supplementary Figure S4.

The smaller effect of stimulus history for $n-1$ incorrect trials was discussed in the previous Reply letter:

...However, an additional factor might be at work. The sensory-perceptual process on trial n might be conditioned by a non-veridical sensory representation after an incorrect (unrewarded) trial $n-1$. If stimulus $n-1$ was weak but incorrectly judged as strong, why did the error occur? It is possible that $n-1$ was erroneously encoded by the sensory system as stronger than its mean expected value (due to noise or fluctuations); the trial n representation would then be less repulsed from weak as a consequence of $n-1$ being erroneously encoded as strong.

This and related issues are addressed more robustly and in detail with Fig. 3c and the related GLM in Supplementary Table T1.

Reviewer #3

2. In addition, what is the accuracy of the GLM included in the response, i.e., to what extent can the GLM predict behavioral outcome.

Authors' Reply

The GLM accuracy, when setting the threshold for “strong” response at 0.5 probability, is 76.44%. Considering the performance of a model that has only information about the current stimulus category (i.e. the performance of the rats), this is 75.98%. Adding the effects of previous stimulus and reward accounts therefore for slightly more than 0.4% accuracy. When we look at trials in which $\Delta speed = 0$, the accuracy is 55.58%, signifying an improvement of almost 6%.

Reviewer #3

3. The last sentence of the response ‘In short, we think that highly trained rats are no longer “learning” in the strict sense of the term’, possibly contradict some earlier results: 1) Well-trained rats improve their performance over time with each session (Fig. 2), and 2) Some of them can learn a variant of the task (Fig. 2).

Authors' Reply

Yes, this point is well-taken. The term “learning” can have different meanings in different contexts and we apologize for the imprecision. Since this phrase was only in the Reply, not in the actual manuscript, we have not implemented any change in the resubmission.

Reviewer #3

Response to point #10: Fig. S4a, x axis in the right two panels seems incomplete?

Fig. S5a, b, no labels for the y axis in the right panels?

Authors' Reply

These details have been corrected. Note that in the new revision, the supplementary figures are renumbered.

Reviewers' Comments:

Reviewer #2:

Remarks to the Author:

I thank the author for the inclusion of Supplementary Fig. S9, and their detailed explanation of the human choice history effect. My concerns have now been adequately addressed and I am happy to recommend this manuscript for publication.

It is curious to see that, in contrast to previous studies in humans, choice repetition increases with increasing ITI, but I concur with the authors that the different timescales may preclude direct comparisons to these previous studies. I am looking forward to reading their future work disentangling these effects.

Reviewer #3:

Remarks to the Author:

The authors have addressed all of the raised questions.